# Dense, continuous membrane labeling and expansion microscopy visualization of ultrastructure in tissues

Tay Won Shin[1,2,3], Hao Wang [1,4,16], Chi Zhang[1,16], Bobae An[1], Yangning Lu[1], Elizabeth Zhang [1], Xiaotang Lu [5], Emmanouil D. Karagiannis[1], Jeong Seuk Kang[1], Amauche Emenari[1,2], Panagiotis Symvoulidis [1], Shoh Asano[1,13], Leanne Lin [6], Emma K. Costa[2,14], IMAXT Grand Challenge Consortium*, Adam H. Marblestone[1,15], Narayanan Kasthuri[7,8], Li-Huei Tsai [2,4] & Edward S. Boyden [1,2,3,6,9,10,11,12] ✉

Lipid membranes are key to the nanoscale compartmentalization of biological systems, but fluorescent visualization of them in intact tissues, with nanoscale precision, is challenging to do with high labeling density. Here, we report ultrastructural membrane expansion microscopy (umExM), which combines an innovative membrane label and optimized expansion microscopy protocol, to support dense labeling of membranes in tissues for nanoscale visualization. We validate the high signal-to-background ratio, and uniformity and continuity, of umExM membrane labeling in brain slices, which supports the imaging of membranes and proteins at a resolution of ~60 nm on a confocal microscope. We demonstrate the utility of umExM for the segmentation and tracing of neuronal processes, such as axons, in mouse brain tissue. Combining umExM with optical fluctuation imaging, or iterating the expansion process, yields ~35 nm resolution imaging, pointing towards the potential for electron microscopy resolution visualization of brain membranes on ordinary light microscopes.

Expansion microscopy (ExM)[1] physically magnifies preserved biological specimens by covalently anchoring biomolecules and/or their labels to a swellable polymer network (such as sodium polyacrylate) synthesized in situ throughout a specimen, followed by chemical softening of the sample, and the addition of water to swell the hydrogel. As the hydrogel swells, the anchored biomolecules and/or their labels are pulled apart from each other isotropically, typically to a physical magnification of ~4-10x in each linear dimension. With an

[1]McGovern Institute, Massachusetts Institute of Technology, Cambridge, MA 02139, USA. [2]Department of Brain and Cognitive Sciences, Massachusetts Institute of Technology, Cambridge, MA 02139, USA. [3]Department of Media Arts and Sciences, Massachusetts Institute of Technology, Cambridge, MA 02140, USA. [4]Picower Institute for Learning and Memory, Massachusetts Institute of Technology, Cambridge, MA 02139, USA. [5]Department of Cellular and Molecular Biology, Harvard University, Cambridge, MA 02138, USA. [6]Department of Biological Engineering, Massachusetts Institute of Technology, Cambridge, MA 02139, USA. [7]Center for Nanoscale Materials, Argonne National Laboratory, Lemont, IL 60439, USA. [8]Department of Neurobiology, University of Chicago, Chicago, IL 60637, USA. [9]Koch Institute, Massachusetts Institute of Technology, Cambridge, MA 02139, USA. [10]Center for Neurobiological Engineering, Massachusetts Institute of Technology, Cambridge, MA 02139, USA. [11]K. Lisa Yang Center for Bionics, Cambridge, MA 02139, USA. [12]Howard Hughes Medical Institute, Cambridge, MA 02139, USA. [13]Present address: Pfizer Inc, Cambridge, MA 02139, USA. [14]Present address: Department of Neurology and Neurological Sciences, Stanford University School of Medicine, Stanford University, Stanford, CA 94305, USA. [15]Present address: Convergent Research, Cambridge, MA 02140, USA. [16]These authors contributed equally: Hao Wang, Chi Zhang. *A list of authors and their affiliations appears at the end of the paper. ✉e-mail: edboyden@mit.edu

iterative form of ExM[2,3], the expanded sample can be expanded a second time, resulting in an overall physical magnification of beyond 10x in each linear dimension. The net result of the expansion is that biomolecules and/or their labels that are initially localized within the diffraction limit of a traditional optical microscope can now be separated in space to distances far enough to resolve them. Expansion microscopy protocols are increasingly prevalent in biology for visualizing proteins[4–8], nucleic acids[9–11], and membrane or lipids[6–8,11–16]. ExM also enables the visualization of anatomical features of specimens through dense labeling of total protein[4,7,8] via N-hydroxyl succinimide (NHS) ester staining. While several ExM methods have been reported for membrane or lipid labeling and visualization[6–8,11–16], achieving dense labeling in fixed tissues has remained challenging (Supplementary Table 1). Ideally, one would enable uniform and continuous membrane labeling, yielding a high signal-to-background ratio, in order to preserve ultrastructure alongside visualization of associated proteins, in fixed tissues. Such a membrane labeling method would enable not just imaging of proteins with nanoscale registration relevant to membrane landmarks, but facilitate the segmentation and tracing of membranous structures, such as axons and dendrites, on a confocal microscope.

Here, we report a strategy to achieve the set of features described above. We rationally and systematically designed innovative membrane labeling probes, and optimized the ExM protocol, to achieve dense labeling of membranes, including plasma membranes, with ExM. We found that our probe, in the context of ExM, labels plasma membranes, mitochondrial membranes, nuclear membranes, ciliary membranes, myelin sheaths, and extracellular vesicle membranes in fixed mouse brain tissue (Supplementary Table 1). We named our protocol ultrastructural membrane expansion microscopy (umExM), using the word ultrastructure in the same sense as an earlier protocol in the expansion microscopy community, called ultrastructure expansion microscopy[17]. umExM preserves ultrastructure and enables the visualization of membranous structures in 100 μm-thick slices of fixed mouse brain at a resolution of ~60 nm with excellent uniformity and continuity of membrane labeling as well as a high signal-to-background ratio (40-80 fold higher than background). umExM could support co-visualization of membranous structures along with proteins and RNAs. The dense membrane labeling of umExM enabled the segmentation of neuronal compartments (e.g., cell bodies, dendrites, and axons), and tracing of neuronal processes (e.g., axons). Finally, we explored combining umExM with optical fluctuation imaging, as well as an iterative form of umExM that achieves a higher expansion factor, enabling ~35 nm resolution imaging of membranes with a standard confocal microscope. We anticipate umExM to have a variety of uses in neuroscience and biology, for the investigation of ultrastructure, cellular compartments, and molecular content, in intact tissues, with nanoscale precision.

## Results

### Design of ultrastructure membrane expansion microscopy chemistry

To develop a membrane labeling probe that labels membranes densely enough to support nanoscale resolution imaging and allow continuous tracing of membranous structures, with ExM chemistry, we designed an unnatural synthetic amphiphilic membrane labeling probe with the following features. First, the membrane labeling probe should exhibit lipophilicity, similar to traditional fluorescent lipophilic dyes like DiI, to enable its preferential localization and diffusion within membranes[18]. The lipophilic hydrocarbon side chains of DiI, for example, are inserted into the hydrophobic regions of membranes[19]. Second, the membrane labeling probe should have a chemical handle that allows for selective conjugation of fluorophores subsequent to the formation of the ExM polymer. This design ensures that the membrane labeling probe remains small in size, facilitating its diffusion and preventing potential degradation of the fluorophore during free-radical polymerization of

the ExM gel[5]. Third, the membrane labeling probe should have a polymer-anchorable handle to incorporate into the ExM gel network for physical expansion. We reasoned that these three features collectively would enable the development of a membrane labeling probe that achieves both dense membrane coverage and compatibility with ExM chemistry, allowing for nanoscale imaging of membranous structures with a standard confocal microscope.

Our probe design proceeded in two phases – a preliminary phase and a final phase. The preliminary phase was used to explore certain aspects of chemical space, and to validate certain aspects of dense membrane staining in ExM. The final phase was then used to refine the properties of the stain for optimal performance, and to perform an even more detailed validation of the density of the membrane staining possible. We include both phases in this paper, although, since the final stain has better performance than the preliminary one, we have placed the images related to the preliminary stain in the Supplementary Figs., so that the final paper focuses on the reagent of greatest use to the reader – the product of the final phase.

During the preliminary phase, we designed the membrane labeling probe to contain a chain of lysines with primary amines for binding to a polymer-anchorable handle, such as acryloyl–X (AcX)[5], previously used to anchor protein amines to the ExM hydrogel[5]. To achieve membrane labeling, we included a lipid tail on the amine terminus of the lysine chain, with a glycine in between, to provide mechanical flexibility[20]. We chose to use D-lysines, rather than the biologically typical L-lysines, to minimize degradation during the chemical softening step of ExM, which in its most popular form involves a proteinase K softening step[5]. Finally, we attached a chemical handle to the carboxy terminus of the lysine chain, for selective conjugation to fluorophore(s) after expansion. In our preliminary design, we chose to use palmitoyl and biotin as the lipid tail and chemical handle, respectively, and to include five D-lysines in the backbone. This design resulted in a glycine and penta-D-lysine peptidic backbone, with a palmitoyl group on the amine-terminus and a biotin on the carboxy-terminus. We named this preliminary probe pGk5b (palmitoyl-glycine-(D-lysine)$_5$-biotin).

We used electron microscopy (EM) to validate our preliminary probe design (with biotin replaced with azide (and denoted pGk5a) so that gold nanoparticles could be added via click chemistry for EM imaging) and observed that membranes were labeled (see Supplementary Note 1; Supplementary Fig. 1). We applied pGk5b to a standard cell line (HEK 293), performed ExM, imaged with a confocal microscope (unless otherwise noted, we used a spinning disk confocal microscope throughout), and observed labeling of membranes (Supplementary Fig. 2a, b). We call this preliminary method, using pGk5b, membrane expansion microscopy (mExM). We evaluated the isotropy of mExM and expansion factor as commonly done with ExM technologies[1,5] and observed distortion and expansion factor comparable to previous ExM protocols (see Supplementary Note 2; Supplementary Figs. 3, 4). Almost all of the pixels exhibiting reference indicators (e.g., mitochondrial matrix-targeted GFP, which is indistinguishable from mitochondrial membrane after ~4x expansion; it requires ~30 nm resolution to distinguish them[21,22], and ER membrane-targeted GFP), also exhibited pGk5b labeling (see Supplementary Note 3; Supplementary Fig. 5). Thus, mExM could accurately visualize mitochondria and ER in cells. mExM was compatible with slices of fixed mouse brain and provided more details compared to the unexpanded state (Supplementary Fig. 6) and was compatible with antibody staining (see Supplementary Note 4). We applied mExM to fixed brain tissue from mice and performed antibody staining against organelle-specific membrane-localized proteins including TOM20 for mitochondria, NUP98 for the nuclear pore complex, and MBP for myelin, and found in all cases that >98% of the pixels exhibiting these reference indicators also exhibited pGk5b signals (see Supplementary Note 5; Supplementary Fig. 7). These preliminary results suggested that it was possible to

make a label that was capable of supporting low-distortion, high-fidelity (as reflected by organelle reference marker colocalization) membrane staining for ExM tissue processing, but some issues remained – for example, the plasma membrane, key to tracing the boundary of neuronal processes, remained hard to see.

Having finished the preliminary phase of the project, we next sought to optimize mExM further. We compared membrane probes with saturated (palmitoyl[23]) and unsaturated (farnesyl[24]) lipids, while keeping the rest of the probe design constant, and observed that the palmitoylated probe achieved a denser membrane labeling compared to the farnesylated one (Supplementary Fig. 8a, b). Furthermore, using a mixture of palmitoylated and farnesylated probe did not achieve denser membrane labeling (Supplementary Fig. 8c). Omitting the glycine linker caused a loss of detail (Supplementary Fig. 9). Finally, we varied the total number of lysines in the backbone of the membrane labeling probe. We reasoned that having more lysines would increase the positive charge of the probe, which could help promote interactions between the probe and chemically-fixed and negatively-charged membranes[25,26]. To explore this, we prepared a series of probes varying in the number of lysines (i.e., 3, 7, 9, 11, 13, and 19 lysines) in the backbone of the probe while holding other moieties known to be useful (i.e., palmitoyl tail, glycine, and biotin) constant. We applied these probes to slices of fixed mice brain and performed ExM. We observed that the probe containing 13 or more lysines appeared to show the boundary of neuronal processes the best (Supplementary Fig. 10). We finalized upon a probe with 13 lysines (pGk13b) to minimize probe size, to facilitate its diffusion throughout brain tissue.

To confirm whether the probe labels the boundary of neuronal processes, we applied pGk13b to slices of fixed Thy1-YFP mouse brain, which expresses cytosolic yellow fluorescent protein (YFP) under the Thy1 promoter in subsets of neurons[27]. We then performed ExM and fluorescently labeled pGk13b with Cy3-conjugated streptavidin, treating the sample with anti-GFP (many fluorescent proteins survive proteinase K softening[5]; anti-GFP binds YFP) to boost YFP signals. We observed that YFP-filled processes were flanked by pGk13b staining (Supplementary Fig. 11), confirming the successful visualization of neuronal boundaries (i.e., plasma membranes) with a standard confocal microscope.

We explored using azide as a chemical handle (resulting in a reagent we named pGk13a) instead of biotin (the aforementioned pGk13b) to increase membrane signals in the context of ExM. We reasoned that fluorescently labeled streptavidin, with four biotin-binding sites[28], could potentially crosslink four pGk13b molecules, thus decreasing fluorescent signals compared to a one-to-one labeling chemistry where each membrane probe binds one fluorescent molecule (as in pGk13a). Furthermore, streptavidin may bind to endogenously biotinylated proteins that are in, for example, mitochondria[29]. We compared pGk13b + Cy3-streptavidin (for which each streptavidin bears more than one Cy3, according to the vendor) and pGk13a + Cy3-DBCO (exhibiting one Cy3 per DBCO) in the context of ExM imaging of the hippocampus in fixed mouse brain slices. We found that the mean signal of pGk13a was >2x higher than that of pGk13b (Supplementary Fig. 12). Thus, we finalized our probe as pGk13a (palmitoyl-glycine-(D-lysine)$_{13}$-azide, Fig. 1a), and used it for the rest of the studies.

We reasoned that preserving membrane integrity in the sample is critical for achieving dense labeling of membranes via pGk13a. However, achieving this is not trivial, in part because many lipids[30] are not fixed through standard paraformaldehyde (PFA) chemical fixation[31]. To better preserve membrane integrity, we added a small amount (0.5%) of calcium chloride (CaCl$_2$, known to help with preserving plasma membranes[32,33]) to 4% PFA fixative. In addition, we maintained a consistent temperature of 4 °C (a cold temperature at which lipids are more ordered, thus reducing the possibility of them diffusing out of the sample[34]) throughout tissue processing until the completion of ExM gel formation, to mitigate potential lipid loss, as higher

temperatures can exacerbate this process[35]. To assess whether this was helpful, we prepared brain slices from mice that were fixed with 4% PFA and 0.5% CaCl$_2$ at 4 °C, and performed standard ExM (37 °C gelation) or modified ExM procedure (4 °C gelation), and imaged hippocampal regions with a confocal microscope, finding that the mean signal of pGk13a from the modified ExM procedure (4 °C gelation) was ~50% higher than from the standard ExM procedure (37 °C gelation) (Supplementary Fig. 13). Thus, we finalized our protocol as follows: we fixed the mouse brain in 4% PFA and 0.5% CaCl$_2$, sectioned the brain, quenched excess aldehydes with a commonly used 100 mM glycine 1x phosphate-buffered saline (PBS) solution, applied pGk13a at 150 μM, applied a previously established biomolecule anchoring solution (acrylic acid N-hydroxy succinimide ester (AX, a reagent that is smaller, more cost-effective, yet functionally analogous to AcX[5] in the context of ExM.) in MES buffer, pH 6.0)[36] to the pGk13a labeled tissue, and finally cast the expandable hydrogel in the tissue − all at 4 °C. We then softened the sample with proteinase K softening solution[1,5], fluorescently labeled pGk13a via click-chemistry, and expanded the sample with water. We named this protocol, using a finalized probe (pGk13a, Fig. 1a) and optimized ExM protocol (Fig. 1b), as umExM (ultrastructure membrane expansion microscopy).

## Validation of umExM

We evaluated the isotropy of umExM by quantitatively comparing pre-expansion structured illumination microscopy (SIM) images to post-expansion confocal images of the same sample, and calculating the distortion across the images as we did for mExM above (see Supplementary Note 2; Supplementary Fig. 3a–g). In summary, we imaged fixed cells expressing mitochondria matrix-targeted GFP with SIM, performed umExM and imaged the same cells with a confocal microscope. Comparing pre-expansion SIM images of expressed GFP (Fig. 2a) to post-expansion images of anti-GFP (Fig. 2b), or of pGk13a (Fig. 2c), we observed the same low distortion (Fig. 2d, e) as was found in previous ExM protocols[1,5]. By comparing the distance between two landmarks in pre- vs. post-expansion images of the same sample, the expansion factor could be calculated; we obtained an expansion factor (~4x) similar to what was previously reported[1,5] (Fig. 2f). Finally, we sought to see how DBCO-Cy3 itself might contribute to membrane labeling, as DBCO itself is lipophilic. We observed that umExM without pGk13a staining did not reveal overt staining, when compared to umExM images acquired with pGk13a (Supplementary Fig. 14).

Having established the isotropy of umExM expansion, we next sought to examine whether this was sufficient to resolve known ultrastructural features previously reported using EM or super-resolution microscopy. To explore this, we first measured the effective resolution of umExM via Fourier Ring Correlation (FRC) resolution analysis[37–39], a gold-standard method which uses Fourier transformation of images to measure resolution, on pGk13a signals from expanded samples. We applied umExM to fixed brain slices from mice and imaged the hippocampus, obtaining a resolution of ~60 nm (Fig. 2g) with a 60x, 1.27NA water objective, similar to the previously reported effective resolution of ExM protocols with similar expansion factor[1,5]. To explore ultrastructural features, we applied umExM to fixed brain slices from Thy1-YFP mice and boosted the YFP signals with anti-GFP treatment. We then imaged the hippocampal dentate gyrus (Fig. 3a, b), third ventricle (Fig. 3c), and somatosensory cortex layer (L) 6 (Fig. 3d, e). We identified axons by examining pGk13a signal flanking anti-GFP signals (Fig. 3f). We quantified the diameter of unmyelinated axons (i.e., in the dentate gyrus; Fig. 3f) and myelinated axons (i.e., in the somatosensory cortex; Fig. 3h, i), and found axon diameters comparable to those obtained from the same brain regions imaged with EM[40–42] (see Supplementary note 6). We also identified motile cilia in the third ventricle by their fingerlike morphologies (Fig. 3j); their diameter (Fig. 3k) was comparable to previous measurements made using EM[43]. We imaged a volume of the third ventricle and visualized it

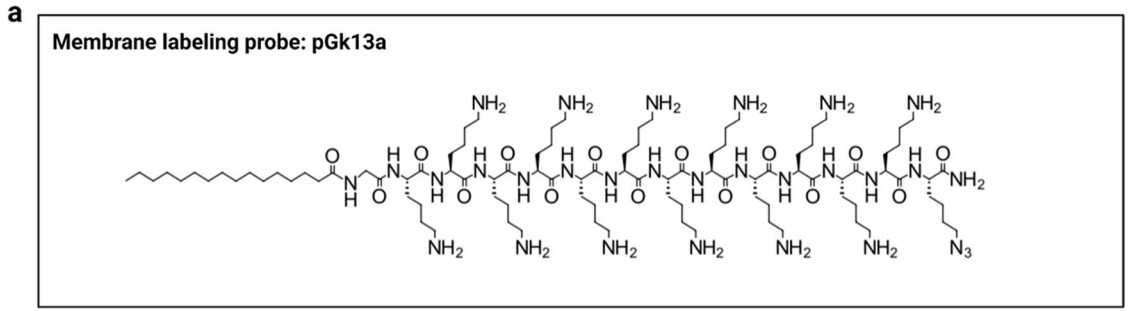

**Fig. 1 | Ultrastructural membrane expansion microscopy (umExM) concept and workflow.** umExM is a modified form of expansion microscopy with a custom-designed amphiphilic membrane labeling probe (termed pGk13a). **a** Chemical structure of pGk13a. The probe does not contain any fluorophore but has an azide to bind a fluorophore later. **b** umExM workflow. Blue-colored fine text highlight key differences from ExM[1] and proExM[5], whereas black fine text highlight the same steps as ExM and proExM. **b. i** A specimen is perfused and chemically fixed with 4% paraformaldehyde (PFA) + 0.5% calcium chloride (CaCl₂) at 4 °C for 24 hours. The brain is sliced on a vibratome to 100 µm thickness at 0-4 °C. **b.ii** The specimen is treated with pGk13a (structure is depicted in (**a**)) at 4 °C overnight (unless otherwise noted, overnight means >16 hours). **b. iii** The specimen is treated with acrylic acid N-hydroxysuccinimide ester (AX) at 4 °C overnight. **b. iv** The specimen is embedded in an expandable hydrogel (made with N,N′-Diallyl-L-tartardiamide (DATD) crosslinker[4]) at 4 °C for at least 24 hours. **b. v** The sample (specimen-embedded hydrogel) is chemically softened with enzymatic cleavage of proteins (i.e., non-specific cleavage with proteinase K) at room temperature (-24 °C) overnight. The probe is not digested during proteinase K treatment since it is composed of D-amino acids. **b. vi** Then, the sample is treated with 1x phosphate-buffered saline (PBS) to partially expand it. The pGk13a, that is anchored to the gel matrix, is fluorescently labeled via click-chemistry (i.e., DBCO-fluorophore) at room temperature, overnight. **b. vii** The sample is expanded with water at room temperature for 1.5 hours (exchanging water every 30 minutes).

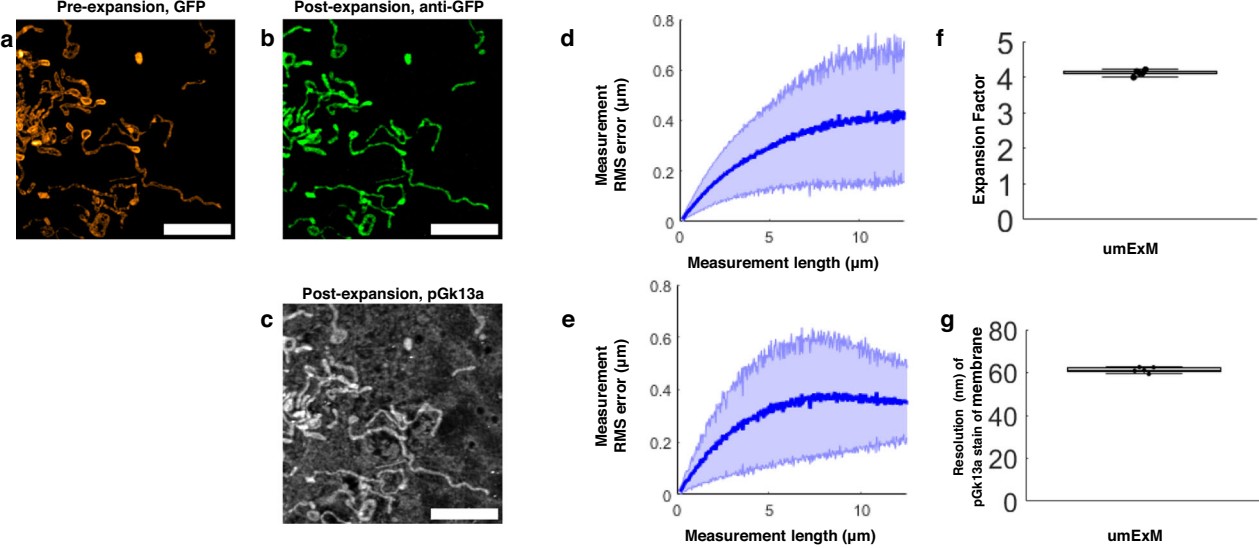

**Fig. 2 | Resolution and distortion of umExM. a** Representative (n = 3 cells from one culture) single z-plane structured illumination microscopy (SIM) image of a pre-expanded HEK293 cell expressing mitochondrial matrix-targeted green fluorescent protein (GFP, shown in orange). **b** Single z-plane confocal image of the same HEK293 cell as in (**a**), after undergoing the umExM protocol, showing expression of mitochondrial matrix-targeted GFP in the same field of view as shown in (**a**). GFP, green color. **c** Single z-plane confocal image of the same umExM-expanded fixed HEK293 cell as in (**a**), showing pGk13a staining of the membrane in the same field of view as shown in (**a**). pGk13a, gray color. **d** Root-mean-square (RMS) length measurement error vs. measurement length, comparing pre-expansion SIM and post-expansion confocal images of cells with mitochondrial matrix-targeted GFP (blue line, mean; shaded area, standard deviation; n = 3 cells). **e** As in (**d**) but with post-

expansion images showing pGk13a staining of the membrane. **f** Boxplot showing measured expansion factor as described (*n* = 4 pairs of landmark points; from 3 fixed brain slices from two mice; median, middle line; 1st quartile, lower box boundary; 3rd quartile, upper box boundary; error bars are the 95% confidence interval; black points, individual data points; used throughout this manuscript unless otherwise noted). **g** Boxplot showing resolution of post-expansion confocal images (60x, 1.27NA objective) of umExM-processed mouse brain tissue slices showing pGk13a staining of the membrane (*n* = 5 fixed brain slices from two mice). Scale bars are provided in biological units throughout all figures (i.e., physical size divided by expansion factor): (**a**–**c**) 5 µm. Source data are provided as a Source Data file.

through 3D volume rendering (Fig. 3l), and found membrane vesicles (known as extracellular vesicles; yellow arrows in Fig. 3l and Supplementary Fig. 15) around cilia, similar to what was previously seen with EM[44,45]. We also imaged the choroid plexus (Supplementary Fig. 16) and observed microvilli, also showing a similar topology to what was previously seen with EM (Fig. 2 from ref. 46).

**Uniform and continuous labeling of membranes by umExM**

We evaluated the uniformity of labeling throughout 3D volumes of umExM-processed slices of mouse brain, using a confocal microscope. We investigated variation in overall labeling, as quantified by the average signal-to-background ratio (S/B; pGk13a signal divided by the background; background was calculated as the average of images of empty gel regions) of each XY plane, at different depths in the expanded tissue volume. We applied umExM to a 100 µm thick fixed coronal slice of mouse brain (Supplementary Fig. 17a) and performed large-scan imaging of the expanded sample with a low magnification objective (4x, 0.2NA; Supplementary Fig. 17b) at 30 milliseconds (ms) laser exposure time (see Supplementary Table 4 for details). We then imaged a volume (i.e., entire depth, from z = 0 µm to z = 100 µm with z step size = 0.375 µm, in biological units (that is, divided by the expansion factor) throughout) of a random part of the CA1 region with the same objective at 50 ms laser exposure time for each z-plane (Supplementary Fig. 17c). We then measured the mean S/B ratio of a single z-plane at different depths of the volume and observed a consistently high mean S/B ratio (>40 fold higher than background, Supplementary Fig. 17d) throughout the slice. We then imaged a volume (from z = 0 µm to z = 10 µm, with z step size = 0.125 µm) of the dentate gyrus region of the hippocampus with a high magnification objective (60x, 1.27NA water immersion lens) with 100 ms laser exposure time (Supplementary Fig. 17e). We observed nanoscale features, such as neuronal processes, as we zoomed into the raw dataset (Supplementary Fig. 17f). We performed the same analysis

(as in Supplementary Fig. 17d) and observed consistently high mean S/B (>80 fold higher than background, Supplementary Fig. 17g). Neuronal processes were clearly delineated, when we zoomed into a cross-sectional image of the volume (Supplementary Fig. 17h). We repeated the experiments (n = 3 fixed brain coronal slices from two mice) and observed similar results. With a 60x 1.27NA water immersion objective, we imaged somatosensory cortex (L6, Supplementary Movie 1) and hippocampus (Cornu ammonis (CA) 2, Supplementary Fig. 18, Supplementary Movie 2; dentate gyrus, Supplementary Movie 3). We used 100 µm thick coronal slices of fixed brain from mice and imaged the expanded samples using the same imaging conditions (i.e., 60x lens, 100 ms laser exposure time) throughout the study unless specified otherwise.

We next quantified the continuity of labeled membranes. Specifically, we focused on individual membranes that can be visualized with umExM in the expanded brain samples, such as the ciliary membrane (Fig. 3m–o), which could easily be identified since they are not in close apposition to a second membrane. We observed distinct peaks of pGk13a signals corresponding to ciliary membranes (Fig. 3m). To quantify the continuity of pGk13a labeled membranes, we manually traced ciliary membrane and counted the number of gaps along them, with a gap defined as a region with intensity smaller than two standard deviations below the mean pGk13a signal along the ciliary membrane, that was longer than 60 nm (the effective resolution of umExM using a 60x, 1.27NA water objective; Fig. 2g). We found that >97% of the ciliary membrane was continuous by this metric at a gap measurement length of 60 nm (Fig. 3o).

We then sought to compare umExM to prior commercially available membrane probes used for lipid or membrane imaging, namely BODIPY FL C12, mCling and Biotin-DHPE. We applied these probes to tissue (see Supplementary Note 7), performed ExM, and sought to do the same S/B ratio analysis and continuity analysis as we did above. We

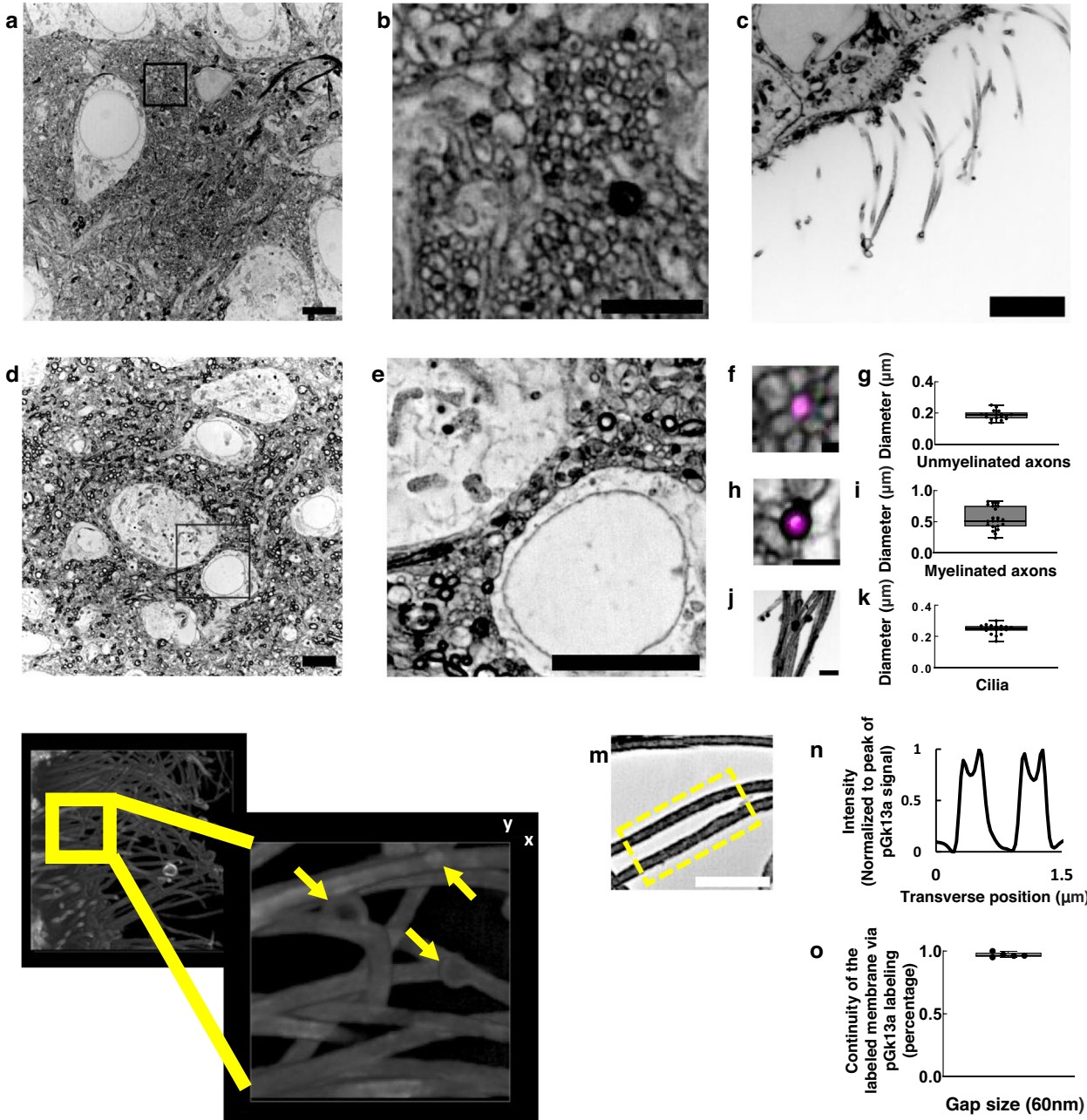

**Fig. 3 | Ultrastructure preservation and continuous labeling of membrane with umExM. a** Representative (n = 5 fixed brain slices from two mice) single z-plane confocal image of expanded Thy1-YFP mouse brain tissue (hippocampus, dentate gyrus) showing pGk13a staining of the membrane. pGk13a staining of the membrane visualized in inverted gray color throughout this figure (dark signals on light background) except for (**l**). **b** Magnified view of black boxed region in (**a**). **c** As in (**a**) but imaging of the third ventricle. **d** As in (**a**) but imaging of mouse somatosensory cortex layer 6 (L6). **e** Magnified view of black boxed region in (**d**). **f** Representative (n = 2 fixed brain slices from two mice) single z-plane confocal image of expanded Thy1-YFP mouse brain tissue (hippocampus dentate gyrus), that underwent umExM protocol and anti-GFP labeling (here labeling YFP), showing YFP (magenta) and pGk13a staining of the membrane (inverted gray). **g** Diameter of unmyelinated axons (n = 17 axons from three fixed brain slices from two mice). **h** As in (**f**), but imaging of somatosensory cortex L6 that was used for measuring the diameter of myelinated axons. (**i**) Diameter of myelinated axons (n = 21 axons from two fixed brain slices from two mice). **j** As in (**f**) but imaging of the third ventricle that was used for measuring the diameter of cilia. **k** Diameter of cilia (n = 19 cilia from two fixed brain slices from two mice). **l** (left) Representative (n = 4 slices of

fixed brains from three mice) volume rendering of epithelial cells in the third ventricle from mouse brain tissue, showing pGk13a staining of the membrane. pGk13a staining of the membrane visualized in gray color. (right) Magnified view of yellow boxed region in (left). Yellow arrows indicate putative extracellular vesicles. Serial image sections that were used for the 3D rendering are in Supplementary Fig. 15. **m** Single z-plane confocal image of expanded mouse brain tissue (third ventricle) processed by umExM, showing pGk5b staining (gray), focusing on the plasma membrane of cilia (i.e., ciliary membrane). **n** Transverse profile of cilia in the yellow dotted boxed region in (**m**) after averaging down the long axis of the box and then normalizing to the peak of pGk13a signal. **o** Boxplot showing the percent continuity of the membrane label (n = 5 separate cilia from two fixed brain slices from one mouse), where we define a gap as a region larger than the resolution of the images (~60 nm, from Fig. 2g), over which the pGk13a signal was two standard deviations below the mean of the intensity of pGk13a along the ciliary membrane. **a** 5 μm, **b** 2 μm, **c** 5 μm, **d** 5 μm, **e** 5 μm, **f** 0.25 μm **h**, **j** 1 μm, (**l**, left) (x); 13.57 μm (y); and 7.5 μm (z) (**l**, right) 3.76 μm (x); 3.76 μm (y); 1.5 μm (z) (**m**) 2 μm. Source data are provided as a Source Data file.

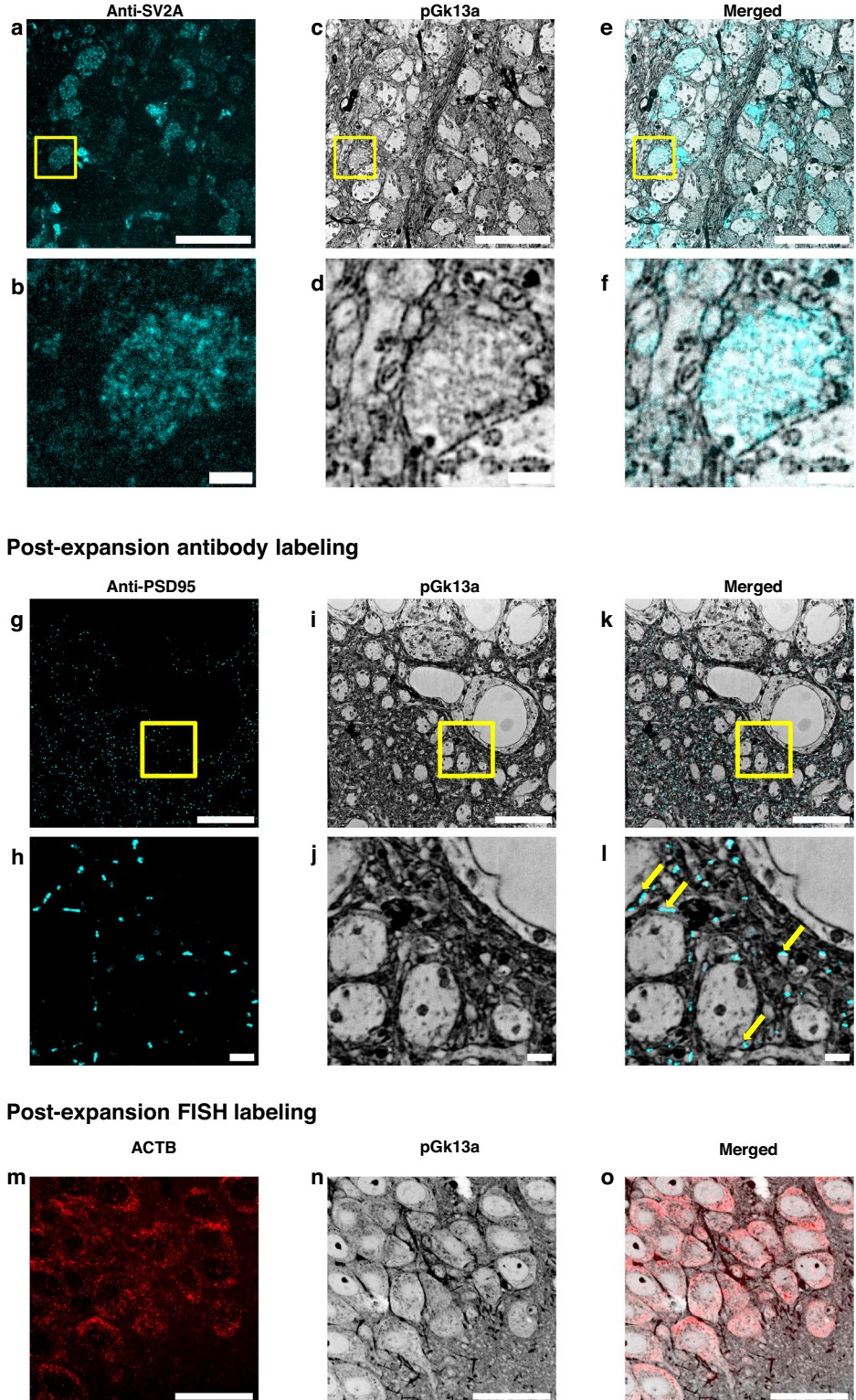

**Pre-expansion antibody labeling**

Anti-SV2A | pGk13a | Merged

**Post-expansion antibody labeling**

Anti-PSD95 | pGk13a | Merged

**Post-expansion FISH labeling**

ACTB | pGk13a | Merged

found that the S/B ratio for umExM images was ~30-39 times higher than that of Biotin-DHPE, BODIPY FL C12, and mCling images (see Supplementary Note 8; Supplementary Fig. 19a–c). We also found that the signals of existing membrane probes were not dense enough to trace ciliary membranes, thus the aforementioned continuity analysis was not possible to perform (see Supplementary Note 9; Supplementary Fig. 19f, g).

## Visualization of proteins and RNAs with umExM

To explore the compatibility of umExM with antibody staining of endogenous protein epitopes, we adopted previously established antibody labeling strategies for ExM, namely pre-expansion antibody staining[5] and post-expansion antibody staining[3]. Notably, post-expansion staining can reveal previously unknown proteins and even cellular structures[3], as antibodies are applied to expanded samples,

**Fig. 4 | umExM with antibody staining and RNA fluorescence in situ hybridization (FISH). a** Representative (n = 5 slices of fixed brain from two mice) single z-plane confocal image of expanded mouse brain tissue (hippocampus, CA3) after umExM processing with a pre-expansion antibody staining protocol (Supplementary Fig. 20), showing immunostaining with an antibody against the synaptic vesicle protein SV2A. **b** Magnified view of the yellow box in (**a**). **c** Single z-plane confocal image of the specimen of (**a**), showing pGk13a staining of the same field of view as in (**a**). pGk13a staining of the membrane visualized in inverted gray color throughout this figure. **d** Magnified view of the yellow box in (**c**). **e** Overlay of (**a**) and (**c**). **f** Magnified view of the yellow box in (**e**). **g** Representative (n = 5 slices of fixed brain from two mice) single z-plane confocal image of expanded mouse brain tissue (hippocampus, CA1) after umExM processing with a post-expansion antibody staining protocol (Supplementary Fig. 21), showing immunostaining against the post-synaptic density protein PSD-95. **h** Magnified view of the yellow box in (**g**). **i** Single z-plane confocal image of the specimen of (**g**), showing pGk13a staining of the same field of view as in (**g**). **j** Magnified view of the yellow box in (**i**). **k** Overlay of (**g**) and (**i**). **l** magnified view of the yellow box in (**c**). The examples of PSD95 signals that were aligned with pGk13a signals were pinpointed with yellow arrows. **m** Representative (n = 3 slices of fixed brain from one mouse) single z-plane confocal image of expanded mouse brain tissue (hippocampus, CA1) after umExM processing with a FISH protocol (Supplementary Fig. 23), showing HCR-FISH targeting ACTB. **n** Single z-plane confocal image of the specimen of (**j**), showing pGk13a staining of the same field of view as in (**j**). **o** Overlay of (**m**) and (**n**). Scale bars: (**a**–**c**, **g**, **h**, **j**) 5 μm, (**d**–**f**, **j**–**l**) 1 μm, (**m**–**o**) 20 μm.

where densely packed proteins are decrowded, making more room for antibody staining.

For umExM with pre-expansion antibody staining, we used a small amount of detergent (i.e., 0.005%-0.01% of saponin or triton-x) to permeabilize membranes in slices of fixed mouse brain tissue, incubated slices with primary antibody at 4 °C, performed umExM, and then incubated the expanded sample with a secondary antibody. Using this protocol (Supplementary Fig. 20), we performed umExM with pre-expansion antibody staining against SV2A, a synaptic vesicle marker (Fig. 4a–f). We found regions of SV2A presence (Fig. 4a, b) in hippocampal area CA3. These signals exhibited pGk13a signals (Fig. 4c, d), consistent with these signals being from synaptic vesicles (Fig. 4e, f).

For umExM with post-expansion antibody staining, we adapted a previous softening method[5,11] that enabled antibody staining after expansion. In particular, we used a softening solution that contained site-specific proteases including trypsin and LysC, and then performed immunostaining after sample expansion. Using this protocol (Supplementary Fig. 21), we performed umExM with post-expansion antibody staining, using an antibody against PSD95 (Fig. 4g–l). We observed a PSD95 expression pattern (Fig. 4g–l) similar to previous post-expansion antibody labeling of PSD95[3]. These signals were adjacent to pGk13a signals (Fig. 4l, yellow arrows), consistent with the known role of PSD95 as a postsynaptic density protein.

To explore whether umExM is compatible with RNA visualization, we combined umExM and ExM visualization of RNA (ExFISH). In particular, we added an RNA anchoring step to the umExM protocol using a previously established RNA anchor (i.e., LabelX)[9], so that the protocol became as follows: we applied pGk13a to label the membrane, applied LabelX anchoring solution followed by AX anchoring solution, and gelled, all at 4 °C. We then softened the tissue with proteinase K softening solution, fluorescently labeled pGk13a, and labeled RNAs with a standard FISH hybridization chain reaction (HCR) protocol. Note that we investigated the use of glycidyl methacrylate (GMA)[11], a previously established reagent for anchoring proteins and RNAs. However, we observed suboptimal membrane visualization after expansion (Supplementary Fig. 22), suggesting the need for separate optimization in this regard. Therefore, we have chosen to move forward with LabelX as the RNA anchor for umExM. We used this protocol (Supplementary Fig. 23) to target ACTB mRNA in fixed brain slices (Fig. 4m–o; used 40x lens). We observed similar gene expression (ACTB) patterns (Fig. 4m) as with the earlier ExFISH protocol[9–11]. Thus, umExM enables simultaneous visualization of membranous structures along with proteins and RNAs, with a standard confocal microscope.

## Segmenting neuron compartments with umExM

We next investigated whether umExM could support the segmentation of neuronal compartments (i.e., cell bodies, dendrites, axons) to help with the analysis of signaling proteins within distinct neuronal compartments. As umExM provides ~60 nm lateral resolution (Fig. 2g), we reasoned that umExM images could capture neuronal processes that are larger than roughly >120 nm (resolution of umExM multiplied by two). To explore this, we applied umExM to fixed brain slices from

Thy1-YFP mice, performed anti-GFP staining to boost YFP signals, and imaged volumes of random regions of somatosensory cortex L6 and hippocampal dentate gyrus. We randomly selected cell bodies using anti-GFP signals, manually segmented them based on pGk13a signals, and then segmented the same cell body based on anti-GFP signals (Fig. 5a) through the commonly used EM image segmentation software, ITK-SNAP[47]. We repeated this procedure for dendrites (Fig. 5b), myelinated axons (Fig. 5c), and unmyelinated axons (Fig. 5d, see **Methods** for details). In summary, we randomly selected dendrites and unmyelinated axons using anti-GFP signals, and for myelinated axons, we employed both anti-GFP and pGk13a signals. This combination was necessary because anti-GFP signals alone could not precisely identify myelinated axons, whereas pGk13a signals were effective in pinpointing them (i.e., strong and thick pGk13a signals due to myelin sheaths; Fig. 3h). Qualitatively, the morphologies of pGk13a signal-guided segmentations were very similar to anti-GFP signal-guided segmentations (Fig. 5a–d). To quantitatively evaluate the accuracy of pGk13a signal-guided segmentation, we utilized the Rand score, a recommended and commonly used metric for assessing EM-based imaging segmentations[48,49], with a Rand score of 0 meaning no similarity between the pGk13a signal-guided versus anti-GFP signal-guided segmentations, and a Rand score of 1 meaning segmentations from the two signals are identical. We observed that pGk13a signal-guided segmentation achieved Rand scores of $0.988 \pm 0.015$ (n = 3 cell bodies from two fixed brain slices from two mice), $0.940 \pm 0.004$ (n = 3 dendrites from two fixed brain slices from two mice), $0.946 \pm 0.013$ (n = 5 myelinated axon from two fixed brain slices from two mice), and $0.890 \pm 0.053$ (n = 5 unmyelinated axon from two fixed brain slices from two mice) for cell bodies, dendrites, myelinated axons, and unmyelinated axons, respectively (Fig. 5e). Although we found that umExM images can capture and support the segmentation of neuronal compartments, thin processes such as tiny axons (as they can be ~50 nm in diameter[50]) and spine necks (known to be ~40–50 nm in diameter[50]) cannot be yet resolved, as umExM provides ~60 nm resolution (Fig. 2g). However, umExM still enables capture and segmentation of neuron compartments that are larger, in fixed tissue, with a standard confocal microscope.

## Tracing axons with umExM

We next sought to explore manual axon tracing supported by umExM images. To explore this, we prepared umExM samples, imaged volumes of expanded samples, and randomly selected myelinated and unmyelinated axons as described above. We first traced pGk13a signals of myelinated axons across the entire image stack (from z = 0 to z = 10.5 μm; Fig. 6a, column "pGk13a") by annotating the centroids of axons in the stacks using the same segmentation software as above (see **Methods** for details; in summary, we used brush size=8 and manually annotated through the stacks). We then repeated the tracing using the anti-GFP signals (Fig. 6a, column "GFP"). The tracing results based on the pGk13a and anti-GFP signals were visually indistinguishable (Fig. 6b). We calculated the Rand score, the same evaluation metric as we used above, and obtained $0.995 \pm 0.004$ (n = 3 myelinated

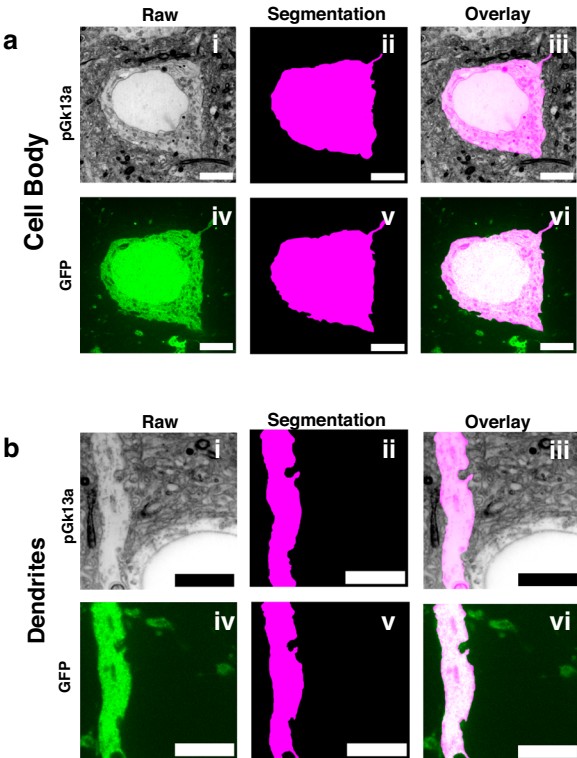

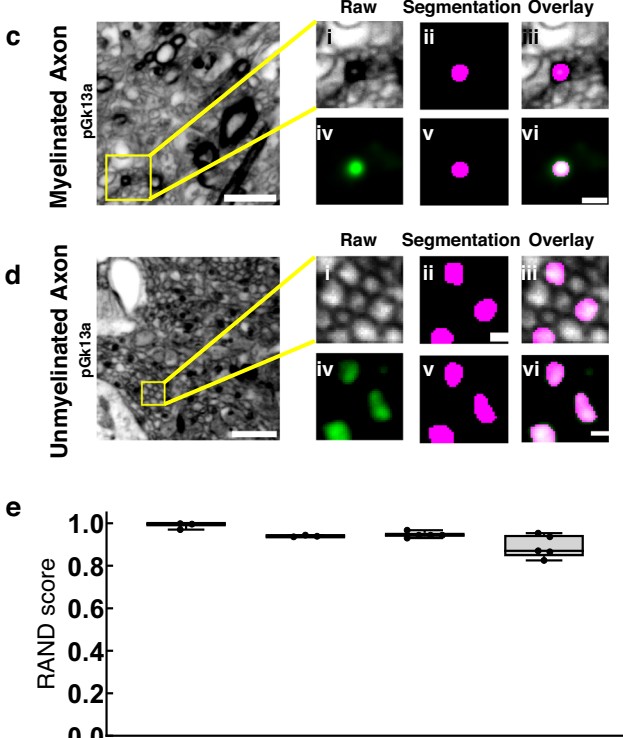

**Fig. 5 | Segmentation ability of umExM. a.i** Single z-plane confocal image of expanded Thy1-YFP mouse brain tissue after umExM processing, showing pGk13a staining of the membrane. **a.ii** Single z-plane image showing manual segmentation of the cell body in (**a.i**). **a.iii** Overlay of (**a.i**) and (**a.ii**). (**a.iv**) Single z-plane confocal image of the specimen of (**a.i**), showing GFP signal of the same field of view as in (**a.i**). (**a.v**) single z-plane image showing manual segmentation of the cell body from (**a.iv**). (**a.vi**) overlay of (**a.iv**) and (**a.v**). **b** As in (**a**), but for segmenting dendrites. **c** (left) Single z-plane confocal image of expanded Thy1-YFP mouse brain tissue showing pGk13a staining of the membrane. (**c.i**) Magnified view of the yellow box on the left. **c.ii** single z-plane image showing manual segmentation of the myelinated axon in (**c.i**). **c.iii** overlay of (**c.i**) and (**c.ii**). **c.iv** Single z-plane confocal image of the specimen of (**c.i**), showing GFP signal of the same field of view as in (**c.i**). (**c.v**) Single z-plane image showing manual segmentation of the myelinated axon in (**c.iv**). (**c.vi**) Overlay of (**c.iv**) and (**c.v**). **d** As in (**c**), but for segmenting unmyelinated axons. (**e**) Rand score of pGk13a signal-guided segmentation of cell body, dendrites, myelinated axon and unmyelinated axons, using anti-GFP signal-guided segmentation as a "ground truth." (n = 3 cell bodies and $n$ = 3 dendrites from two fixed brain slices from two mice, and n = 5 myelinated axons and $n$ = 5 unmyelinated axons from two fixed brain slices from two mice). Scale bars: (**a.i–vi**) 5 μm, (**b.i–vi**) 5 μm, (**c**) (left) 2 μm; (**i–vi**) 0.5 μm, (**d**) (left) 2 μm; (**i–vi**) 0.5 μm. Source data are provided as a Source Data file.

axons from two fixed brain slices from two mice) when we used anti-GFP-guided tracing as the 'ground truth'. We repeated this procedure for unmyelinated axons in the dentate gyrus (Fig. 3f), and obtained 0.993 ± 0.006 (from z = 0 to z = 5.0 μm; Fig. 6b, n = 3 myelinated axons from two fixed brain slices from two mice). However, due to the axial resolution of umExM, which is ~125 nm (axial resolution of a confocal microscope divided by expansion factor; ~500 nm/4) in principle, tracing unmyelinated axons with pGk13a signals alone posed a limitation beyond z = ~5 μm (on average, n = 3).

Next, we imaged the corpus callosum, a brain region containing densely packed myelinated axons. However, we found that manual tracing of neuronal processes was challenging in this region as only a subset of the processes were visually distinguishable (Supplementary Fig. 24a, b), perhaps due to light scattering; this optical phenomenon was not observed in the somatosensory cortex and hippocampus (Supplementary Movies 1–3). Previous studies reported that a subset of native lipids, which causes scattering, may still remain even after tissue clearing[51] and expansion processes[8]. We found that transferring pGk13a and biomolecules to an ExM gel matrix formed post-expansion, and then chemically cleaving the initial ExM gel, exhibited improved visualization of axons in this brain region (Supplementary Fig. 24c, d). In more detail, we performed umExM on a fixed brain slice until the softening step was completed, and then we applied biomolecule anchoring (AX) solution again (so that pGk13 probes in the initial gel could be transferred from the initial gel to a subsequently formed

ExM gel; the newly applied AX would react to unreacted amines in pGk13a), cast an expandable gel that was prepared with non-cleavable crosslinker N,N-methylenebis(acrylamide) (BIS) in the initial gel, chemically cleaved the initial gel (which was made with cleavable crosslinker N,N'-Diallyl-L-tartardiamide (DATD)), fluorescently labeled pGk13a via click chemistry, and expanded the sample with water. Using this protocol (Supplementary Fig. 25), we imaged corpus callosum covering a volume of 39.25 by 39.25 by 20 μm, at 50ms laser exposure time for each single z section. When we zoomed into the dataset, we were able to clearly identify neuronal processes in the corpus callosum (Supplementary Movies 4, 5), similar to what we observed in other brain regions such as cortex and hippocampus (Supplementary Movies 1–3). We used this dataset (**from** Supplementary Movie 5) to manually trace 20 axons in the bundle of myelinated axons (Fig. 6g–i) that spanned the entire dataset without any challenges.

## Higher resolution imaging with umExM

ExM can support higher resolution imaging, by imaging ExM-processed samples with other super-resolution imaging methods[8,52], or by expanding beyond 4 times, e.g. through iterative forms of ExM[2–4]. We explored both of these possibilities. We first combined umExM with an existing super-resolution imaging method. Inspired by recent progress in optical fluctuation imaging with ordinary confocal microscopes[53–55], we chose "super-resolution imaging based on auto-correlation with two-step deconvolution" (i.e., SACD)[53], as this method

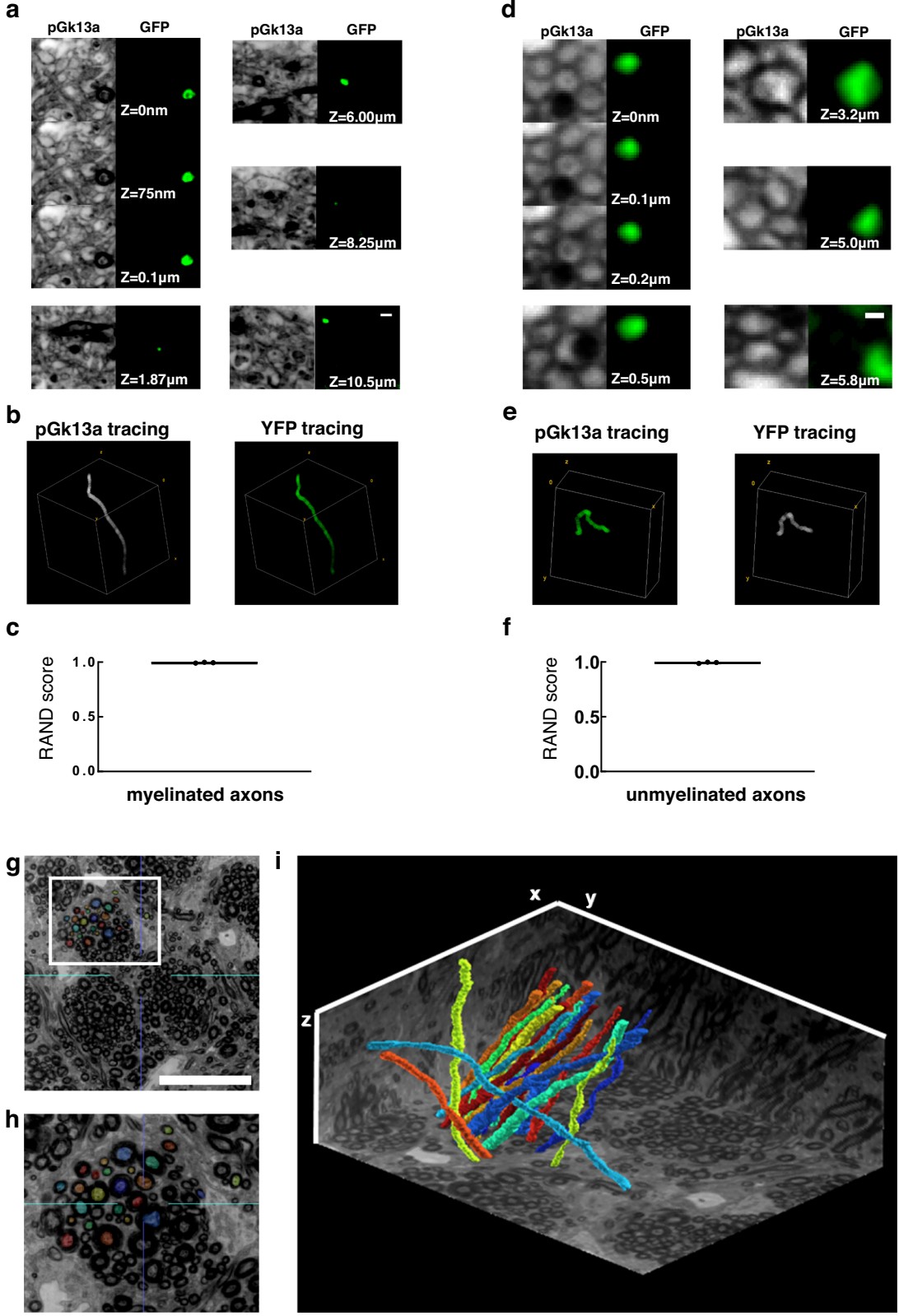

requires fewer frames to resolve fluctuations compared to other methods[53]. We performed umExM of fixed mouse brain slices and used a confocal microscope to image 20 frames of a hippocampal region at an imaging rate of 50 ms/frame (Fig. 7a). We then used the SACD algorithm[53] to resolve the fluctuations (Fig. 7b). We measured the resolution of the resulting image the same way as we did for umExM. umExM+SACD provided a final effective resolution of ~33 nm (Fig. 7c).

We next explored creating an iterative form of umExM, adapted from the previously established iterative form of ExM (iExM)[2]. We performed umExM on fixed brain slices but without fluorescently labeling pGk13a. We then embedded the expanded sample into a re-embedding gel (uncharged gel) prepared with a cleavable crosslinker (DATD) to preserve the expanded state during subsequent steps[2], treated the specimen with biomolecule anchoring (AX) solution again so that the

**Fig. 6 | Traceability of umExM. a** (pGk13a column) Serial confocal images of expanded Thy1-YFP mouse brain tissue after umExM processing, showing pGk13a staining of the membrane. (GFP column) anti-GFP signal of the same sample in the same field of view. **b** (left) pGk13a signal-guided manually traced and reconstructed myelinated axon from (**a**, pGk13a column). (right) As in (left), but with anti-GFP signals. **c** Rand score (n = 3 myelinated axons from two fixed brain slices from two mice) of pGk13a signal-guided manual tracing of myelinated axons, using anti-GFP signal-guided tracing as a "ground truth." **d** As in (**a**) but with an unmyelinated axon. **e** As in (**b**) but for (**d**). **f** As in (**c**) but for unmyelinated axons (n = 3 unmyelinated axons from two fixed brain slices from two mice). **g** Representative (n = 4 fixed brain slices from two mice) single z-plane confocal image of expanded mouse brain tissue (corpus callosum) after umExM with double gelation processing (Supplementary Fig. 25), showing pGk13a staining of the membrane. The seeding points for manual segmentation are labeled with colors. **h** Magnified view of the white box in (**g**). **i** 3D rendering of 20 manually traced and reconstructed myelinated axons in the corpus callosum. Planes were visualized from raw umExM images that were used for tracing. Scale bars: (**a**) 0.5 μm, (**d**) 0.2 μm, (**g**) 18 μm, (**i**) 39.25 μm (x); 39.25 μm (y); and 20 μm (z). Source data are provided as a Source Data file.

pGk13a probes could be transferred from the initial gel to a subsequent gel, cast a new expandable gel prepared with a non-cleavable cross-linker (BIS), chemically cleaved the initial (composed of cleavable crosslinker DATD, as noted above) and re-embedding gels, as in the previously established iterative form of ExM[2,3]. Finally, we fluorescently labeled pGk13a via click chemistry and expanded the sample. Inspired by recent advancements in extracellular space preservation (ECS) fixation[56], we applied this protocol (Supplementary Fig. 26) to ECS-preserved fixed brain slices (Fig. 7d, e, Supplementary Movie 6) and achieved ~12x expansion. We measured the resolution of the resulting image as above, and observed that the iterative form of umExM achieved a final effective resolution of ~35 nm (Fig. 7f). With this protocol, we observed mitochondrial cristae (Supplementary Fig. 27), showing similar appearance as seen with earlier super-resolution imaging methods (e.g., Fig. 2b from ref. 22).

## Discussion

umExM achieves dense labeling of membranes, and high-integrity expansion, to enable imaging of membranous structures using a standard confocal microscope. It achieves ~60 nm lateral resolution and enables co-visualization of membranous structures in a wide range of brain regions along with proteins and RNAs. Although umExM cannot resolve tiny processes such as spine necks, umExM enables segmentation of cell bodies, dendrites, and axons (>200 nm in diameter) and enables tracing of axons. Finally, we showed that ~35nm resolution imaging of membrane structures is possible by combining umExM with super-resolution imaging (e.g., SACD) or through an iterative form of umExM. The cost of pGk13a falls within the price range of commercially available membrane labeling probes used in other ExM technologies (DiD for MAGNIFY[8] and PacSph for panExM-t[7]; Supplementary Table 1). It is worth noting that the cost of pGk13a could decrease greatly with commercial mass production. We tested our protocol on slices 50 to 100 microns thick, but did not test thicker slices in the current study, as we were focused on the chemistry of lipid staining. Expansion microscopy protocols have been extensible to very large samples, including entire mouse brains[57]. Thicker samples may need longer pGk13a incubation times, or a higher concentration of pGk13a, or both, as slice thickness increases.

umExM does not yet have the same resolution as high-end electron microscopy. In addition, the probe, which contains a palmitoyl group, could in principle intercalate differentially with different membrane types. EM-processed samples imaged with low-resolution imaging instruments (e.g., via X-ray imaging, offering ~83nm resolution imaging[58]) can only show membrane-bound objects larger than the resolution. Our 4x protocol visualized mitochondria (Supplementary Fig. 5a–c) and ER (Supplementary Fig. 5d–f), and could reveal some features of cytoplasmic vesicles (i.e., synaptic vesicles; Fig. 4 c–f), although the round shapes of these vesicles could not be seen with the resolution of 4x umExM. The iterative form of umExM, which provides higher resolution (i.e., ~35 nm resolution, Fig. 7f) compared to 4x umExM, revealed mitochondria cristae (Supplementary Fig. 27), with an appearance similar to that shown with isoSTED[22]. We also saw ER-like structures, but we again did not see the round shapes of synaptic vesicles, perhaps due to limited stain density in conjunction with borderline resolution. In umExM images obtained with SACD, we did

not observe mitochondrial cristae, despite the higher resolution; it may be necessary to optimize SACD parameters[53] to see this.

We anticipate that umExM can be effectively combined with other ExM protocols. For instance, we expect that the protocol combining umExM and ExFISH can be simplified by using the universal anchoring reagent GMA[11]. We also expect umExM chemistry can be combined with techniques such as expansion sequencing (ExSeq)[10]. In spatial transcriptomic mapping, cell segmentation heavily relies on the computational extraction of the cell boundary[59]. We expect our technology, which densely labels membranes, to provide helpful information for manual and automatic cell boundary segmentation to facilitate spatial transcriptomic studies.

Future directions may also include further optimizing the iterative form of umExM. Similar to how EM sample processing was optimized by performing thorough and systematic screening of experimental conditions (e.g., concentration and duration of OsO4 staining)[60–63], the iterative form of umExM may be further optimized by systemically tweaking parameters in the protocol (e.g., fixative solution, monomer solution, etc.). Furthermore, one may combine umExM with total protein staining[4,6–8]; this will label unreacted amines in both pGk13a probes as well as proteins, similar to how uranyl acetate staining provides more contrast in the sample for EM imaging. Once the iterative form of umExM is optimized, one could potentially trace neurons and their connectivity, with molecular markers, on a standard confocal microscope.

## Methods
### Membrane probe synthesis
Membrane probes were commercially synthesized (Anaspec). They were purified to >95% purity. They were aliquoted in 1 mg quantities into tubes, lyophilized to powder, and stored at −20 °C until stock solutions were prepared. Stock solutions were stored at −20 °C until use.

### Brain tissue preparation for umExM
All procedures involving animals were in accordance with the US National Institutes of Health Guide for the Care and Use of Laboratory Animals and approved by the Massachusetts Institute of Technology Committee on Animal Care. The animals were kept under standard conditions at a room temperature ~72 °F, with relative humidity at 30–70%, on a 12-hour light/dark cycle. Wild type (both male and female, used without regard to sex, C57BL/6 or Thy1-YFP, 6-8 weeks old, from either Taconic or JAX) mice were first terminally anesthetized with isoflurane. Then, ice-cold 1x phosphate-buffered saline (PBS, Corning, catalog no. 21031CM) was transcardially perfused until the blood cleared (approximately 25 ml). For all umExM experiments, the mice were then transcardially perfused with 4% PFA + 0.5% $CaCl_2$ fixative solution (Supplementary Table 3 "fixative solution"). The fixative was kept on ice during perfusion. After the perfusion step, brains were dissected out, stored in fixative on a shaker (~10−20 rpm) at 4 °C for 24 hours for further fixation, and sliced on a vibratome (Leica VT1000S) at 100 μm thickness. For the slicing, the tray was filled with ice-cold PBS, and the tray was surrounded by ice. The slices were then transferred to a 50-ml tube filled with 40 ml of ice-cold quenching solution (100 mM Glycine in PBS) on the shaker (~10−20 rpm) at 4 °C,

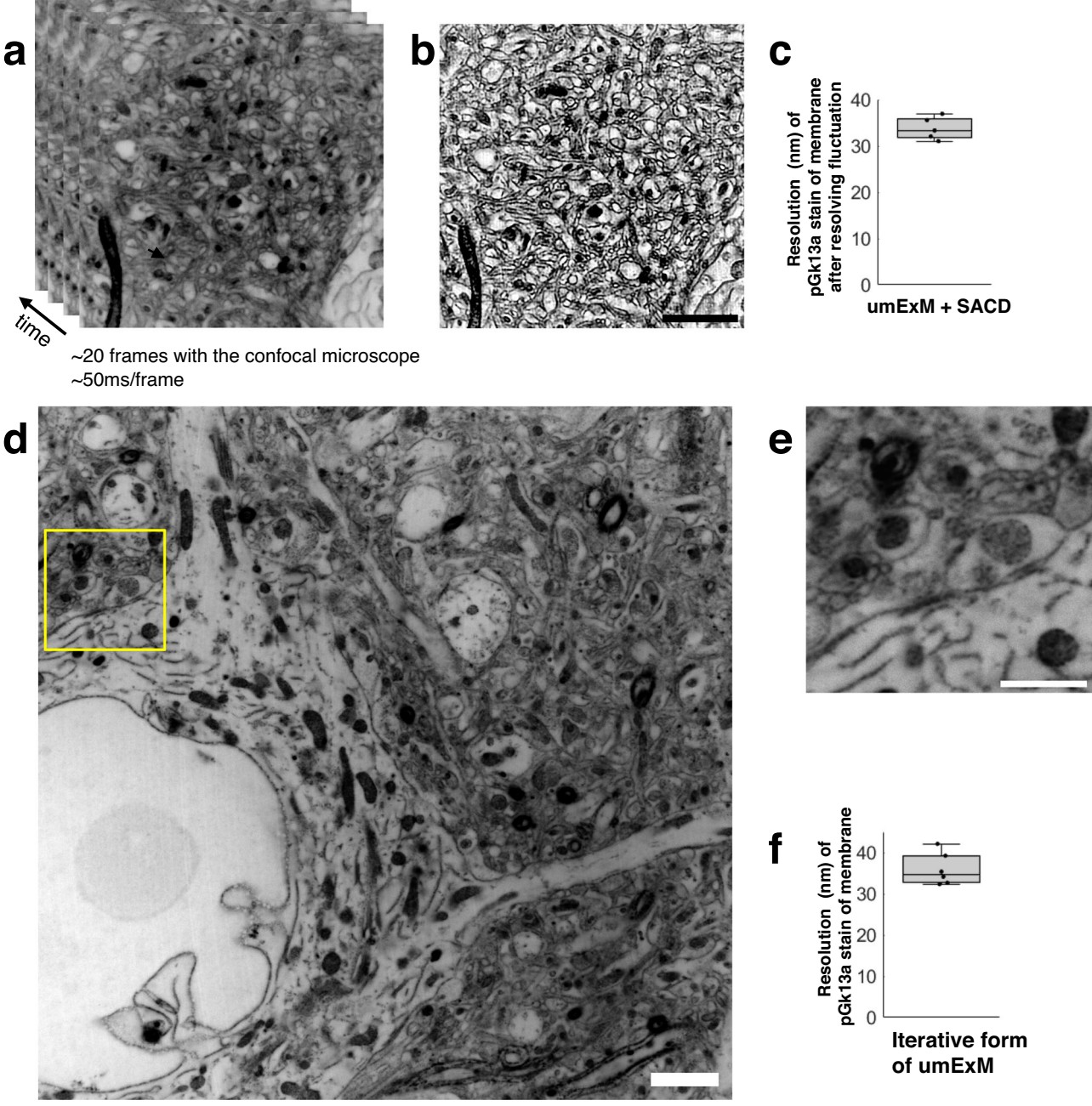

~20 frames with the confocal microscope
~50ms/frame

**Fig. 7 | Higher resolution umExM. a** Representative (n = 5 fixed brain slices from 2 mice) single z-plane confocal image of post-expansion mouse brain tissue (Somatosensory cortex, L4) that underwent the umExM protocol. Images were taken at 50ms/frame for 20 frames with a confocal microscope with 1.5x optical zoom. pGk13a staining of the membrane visualized in inverted gray color throughout this figure. **b** Fluctuations in the acquired frames (as in (**a**)) were resolved with the 'super-resolution imaging based on autocorrelation with a two-step deconvolution' (SACD) algorithm[53]. **c** Boxplot showing resolution of post-expansion confocal images (60x, 1.27NA objective) of umExM + SACD-processed mice brain tissue slices showing pGk13a staining of the membrane (n = 5 fixed brain slices from two mice). **d** Representative (n = 6 fixed brain slices from one mouse) single z-plane confocal image of post-expansion mouse brain tissue (Somatosensory cortex, L4) after the iterative form of umExM processing (Supplementary Fig. 26), showing pGk13a staining of the membrane. **e** Magnified view of yellow box in (**d**). **f** as in (**c**) but for the iterative form of umExM (n = 6 fixed brain slices from one mouse). Scale bars: (**b**) 10 μm, (**d**, **e**) 1.5 μm. Source data are provided as a Source Data file.

overnight (>8 hrs). The slices were washed 3-4 times with ice-cold PBS on the shaker (-10–20 rpm) at 4 °C, for 1–2 hours each and stored in PBS at 4 °C.

**umExM for brain tissue slices**

1. The fixed tissue slices (as described in the **Brain tissue preparation for umExM** section) were incubated in membrane labeling solution (Supplementary Table 3, "pGk13a stock solution") on the shaker (-10-20 rpm) at 4 °C, overnight (unless otherwise noted, overnight means >16 hours).

2. The fixed tissue slices were then incubated in AX stock solution (Supplementary Table 3, "AX stock solution") on the shaker (-10–20 rpm) at 4 °C, overnight. The tissue was then washed 2-3 times in PBS on the shaker (-10–20 rpm) at 4 °C, 1 hour each.

3. The fixed tissue slices were then incubated in gelling solution (Supplementary Table 3, "umExM gelling solution") 30 minutes on the shaker (-10–20 rpm) at 4 °C for pre-gelation incubation.

During this step, the gelation chamber was constructed similarly as previously described[5]. In summary, we placed two spacers (VWR, catalog no. 48368-085) on a microscope slide (VWR, catalog no. 48300-026). The two spacers were separated from each other enough so that the brain tissue slice could be placed in between them. The brain tissue slice was placed between the spacers and sliced with a razor blade (VWR, catalog no. 55411-050) into two equally sized half-coronal sections. We then placed the lid (VWR, catalog no. 87001-918) on top of the spacers as well as the brain tissue slices. We fully filled the empty space between the half-coronal sections and spacers with the gelling solution. The chamber was transferred to a plastic jar with a lid (Fisher Scientific, catalog no. R685025) at 4 °C to initiate free-radical polymerization for >24 hours. Then, the gelation chamber containing the sample (tissue-embedded hydrogel) was taken out.

4. We trimmed the sample with a razor blade (VWR, catalog no. 55411-050) to have two gelled half-coronal sections. We then transferred each gel (each half-coronal section) from the chamber to a 12-well plate (Fisher Scientific, catalog no. FB012928) that contained proteinase K digestion solution (Supplementary Table 3, "umExM Digestion buffer") in the well (2 ml of digestion solution per well per half-coronal section). The gel was then digested at room temperature (RT, 24 °C) on the shaker (50 rpm), overnight. After digestion, the gels were washed 3-4 times in PBS on the shaker (50 rpm) at RT, 30 minutes each.

5. Each sample was labeled with 0.5 ml of Cy3 conjugated DBCO (Cy3 DBCO Click chemistry tools, catalog no. A140-1) buffered in PBS at a concentration of 0.1 mg/ml on the shaker (50 rpm) at RT, overnight. Then, the samples were washed 2-3 times in PBS on the shaker (50 rpm) at room temperature (RT), 30 minutes each. The samples were then transferred to 4 °C, overnight.

6. The samples were placed 2-3 times in excess water on the shaker (50 rpm) at RT for expansion, 30 minutes each.

## Immunohistochemistry-compatible umExM

For pre-expansion antibody staining (Supplementary Fig. 20), we prepared the brain tissue slice as described in step 1 in the **umExM for brain tissue slices** section. We then applied 1ml of permeabilization solution (0.005%-0.01% of saponin (Sigma Aldrich, catalog no. 84510) or triton (Sigma, catalog no. X100), 1% Bovine Serum Albumin (BSA, Sigma Aldrich, catalog no. A3294) in PBS) at 4 °C, overnight. We then added 10 µl of primary antibody, rabbit anti-SV2A (Abcam, catalog no. ab32942), to the permeabilization solution, and then held it for 24 hours at 4 °C on the shaker (50 rpm). Then, the tissues were washed 3-4 times in PBS at 4 °C, 1 hour each. Next, we performed steps 2-5 in the **umExM for brain tissue slices** sections. Subsequently, for each half coronal slice sample, we incubated it in PBS containing primary antibodies, goat anti-rabbit ATTO 647N conjugated secondary antibody (Rockland Immunochemicals, catalog no. 50-194-3924), at a concentration of 5-10 µg/mL at 4 °C for 2-3 days. The samples (tissue-embedded hydrogel) were washed 3-4 times in PBS at RT, 30 minutes each. Finally, we performed step 6 in the **umExM for brain tissue slices** section.

For post-expansion staining (Supplementary Fig. 21), we performed steps 1-3 in the **umExM for brain tissue slices** section. We then performed step 4 in the **umExM for brain tissue slices** section, but with 2 ml of Trypsin+Lys-C softening solution (Supplementary Table 3, "umExM Trypsin+Lys-C softening solution") instead of proteinase K digestion solution, for each half coronal slice sample. We then incubated each half coronal slice sample (tissue-embedded hydrogel) in PBS containing primary antibodies, rabbi anti-PSD95 (Thermo Fisher, catalog no. MA1-046), at a concentration of 10 µg/ml at 4 °C for 2-3 days. The samples were washed 3-4 times in PBS at RT, 30 minutes each. We then performed step 5 in the **umExM for brain tissue slices** section. Subsequently, for each half coronal slice sample, we incubated it in PBS containing primary antibodies, goat anti-rabbit ATTO 647N conjugated secondary antibody (Rockland Immunochemicals, catalog no. 50-194-3924), at a concentration of 5-10 µg/ml at 4 °C for 2-3 days. Then, the samples were washed 3-4 times in PBS at RT, 30 minutes each. Finally, we performed step 6 in the **umExM for brain tissue slices** section.

## Antibody staining of fluorescent proteins for umExM

The expanded samples, after either proteinase K digestion (steps 1-4 in the **mExM for brain tissue slices** section) or Trypsin+Lys-C softening treatment (post-expansion antibody staining protocol in **Immunohistochemistry-compatible umExM** section), were incubated in PBS containing ATTO 647N fluorophore-conjugated nanobody against the green fluorescent protein (GFP, ChromoTek, catalog no. gba647n) or ATTO 488 fluorophore-conjugated nanobody against the green fluorescent protein (GFP, ChromoTek, catalog no. gba488) at a concentration of 10 µg/ml for overnight at 4 °C. The samples were washed 3-4 times in PBS at RT, 30 minutes each. We then performed steps 5 and 6 in the **umExM for brain tissue slices** section.

## umExM with RNA

For umExM with RNA (Supplementary Fig. 23), we prepared brain tissue slices as described in step 1 in the **umExM for brain tissue slices** section. We then incubated the sample into 1mL of LabelX solution[9] (10 µL of AcX (ThermoFisher, catalog no. A20770), 10 mg/ml in DMSO, was reacted with 100 µL of Label-IT Amine Modifying Reagent (Mirus Bio, catalog no. MIR3900), overnight at RT with shaking). We then performed step 2 in the **umExM for brain tissue slices** section, but with the 0.05 mg/ml AX in MES buffer (See Supplementary Table 3, "AX buffer solution") for 24 hours at 4 °C. We then performed steps 3-6 in the **umExM for brain tissue slices** section. Next, we performed the standard FISH hybridization chain reaction (HCR) protocol, similar to earlier ExM protocols that visualized RNAs[9–11]. In particular, we incubated the sample (tissue-embedded hydrogel) with hybridization buffer (10% formamide, 2× SSC) at RT for 0.5-1 h, and applied ACTB probe (Molecular instruments) at 8nM concentration, overnight at 37 °C (buffered in HCR.v3.0 Wash Buffer[64]). We then washed the gel with HCR v3.0 Wash Buffer for 2-3 times at 37 °C followed by another washing with second washing buffer (5x SSC buffer + 0.1% Tween 20) 30 minutes for 4 times at 37 °C, followed by treating the sample with fluorescently (Alexa 647) labeled HCR hairpin amplifiers (1:100) at RT, overnight. Then the samples were washed with 5× SSCT, 20 minutes for 4 times at RT. The samples were expanded (~3x; similar to the expansion factor of ExFISH[9] that used LabelX for anchoring RNAs) with 0.05× SSCT, 10 – 20 minutes each time, 3 times.

## Confocal imaging, deconvolution, and visualization

Confocal images in the main and Supplementary Figs. were obtained on an Andor spinning disk (CSU-W1 Yokogawa) confocal system on a Nikon Eclipse Ti-E inverted microscope body with a Zyla 5.5 camera or a Hamamatsu qCMOS camera. We used a 4x 0.2 NA, 10x 0.45 NA, 40x 1.15 NA, or 60x 1.27 NA lens for all imaging. For large-scan imaging, we imaged with the confocal microscope and then stitched with a shading correction function via the default setting in Nikon element software version 4.0. All confocal images in the main figures were deconvoluted with the Sparse-deconvolution[65] software (version 1.0.3) using the software provided in GitHub (https://github.com/WeisongZhao/Sparse-SIM). Gaussian filter function (sigma=2) in ImageJ (version 1.53q) was applied to all antibody signals (anti-GFP, anti-SV2A and anti-PSD95). The 3D volume renderings of confocal images were generated using the volume viewer or 3D viewer function in ImageJ (version 1.53q). All images were visualized with an auto-scaling function in ImageJ (version 1.53q) except for Supplementary Figs. 12–14, which we

used the same brightness and contrast with ImageJ software to highlight the difference between experimental outcomes.

## Resolution analysis

For the resolution analysis, we adopted blockwise Fourier Ring Correlation (FRC) resolution analysis[39] to measure the resolution of umExM as well as umExM+SACD and the iterative form of umExM. We normalized the pixel size of our umExM, umExM+SACD and iterative umExM images by the expansion factor, so that the resolution would be described in biologically relevant terms. For umExM and the iterative form of umExM images, the same region of umExM samples was imaged twice for independent noise realization. Then we used NanoJ-SQUIRREL Fiji plugin[39] to perform FRC resolution analysis. In the case of umExM+SACD images, we captured 40 frames of umExM images, divided them into two sets of 20 frames by separating odd and even images, and performed SACD (see **umExM with Optical fluctuation imaging section** below) to generate two SACD images (each derived from 20 frames). Subsequently, these two images underwent FRC resolution analysis using the same Fiji plugin. The best FRC value obtained across the blocks in each image pair was used to quantify the resolution of umExM, umExM+SACD and the iterative form of umExM.

## Analysis of the biotin (pGk13b) vs. azide (pGk13a) version of the membrane probe

The brain tissue sections were prepared as described in the **Brain tissue preparation for umExM** section but with 4% PFA (Electron Microscopy Sciences, catalog no. 15710) solution instead of 4% PFA + 0.5% CaCl2 (Supplementary Table 3, "fixative solution"). To compare the biotin version (pGk13b) of the probe with azide version (pGk13a) of the probe, we performed ExM as described in the **umExM for brain tissue slices** section, but with either pGk13b or pGk13a in step 1 and a typical ExM gelation temperature in step3 (pre-gelation 4 °C and gelation at 37 °C). For fluorescently labeling the pGk13b and pGk13a, we used an excessive amount of Cy3-conjugated streptavidin or DBCO for a long time (~2 days at RT) to fluorescently label the membrane probes, as much as possible. In particular, we used 1ml of PBS containing Cy3 conjugated streptavidin (Invitrogen, catalog no. SA1010) at a concentration of 0.1mg/ml for 2 days at RT. For fluorescently labeling pGk13a, we used 1 ml of PBS containing Cy3 conjugated DBCO (Click chemistry tools, catalog no. A140-1) at a concentration of 0.1 mg/ml for 2 days at RT. Both samples were washed 3-4 times in PBS at RT, 30 minutes each, and expanded with water. We then imaged a random region in the hippocampus with the confocal microscope with 10x, 0.45NA objective. We then measured the mean pGk13a and pGk13b signals. We then performed an unpaired two-sided t-test function in RStudio 2021.09.2 + 382 with R version 4.1.2.

## Analysis of 37 °C vs. 4 °C ExM protocols

For control experiments, we performed ExM as described in the **umExM for brain tissue slices** section but with typical ExM gelation temperature (i.e., gelled at 37 °C for 2 hours in step 3). For 4 °C gelation, we performed ExM as described in the **umExM for brain tissue slices** section. We then imaged the samples in a random region in the hippocampus with the confocal microscope with 10x, 0.45NA objective. We then measured the mean pGk13a from each condition and performed an unpaired t-test function in RStudio 2021.09.2 + 382 with R version 4.1.2.

## Signal-to-background analysis

The umExM samples were prepared as described in the **umExM for brain tissue slices** section. To obtain the mean pGk13a signal, we imaged a volume covering the depth from $z = 0 \ \mu m$ to $z = 100 \ \mu m$ with a z-step size of 0.375 μm (in biological units), using a 4x 0.2 NA lens and Zyla 5.5 camera with a 50ms laser exposure time (see Supplementary Table 4 for details) for each z-plane. To obtain the mean background,

we imaged random empty regions in the gel with the same imaging conditions (i.e., 4x lens, Zyla 5.5 camera, 50ms laser exposure time) and averaged them. We then measured the mean signal-to-background (S/B) by dividing the mean pGk13a signal captured in the XY plane by the mean background (i.e., mean pGk13a signal/mean background). We subsequently calculated the mean signal-to-background (S/B) ratio for a single z-plane at various depths within the volume. We repeated this with a 60x, 1.27 NA lens for a volume covering the depth from $z = 0 \ \mu m$ to $z = 10 \ \mu m$, with a z step size = 0.125 μm.

## Continuity of labeled membrane analysis

The umExM samples were prepared as described in the **umExM for brain tissue slices** section. We randomly traced the ciliary membrane (n = 5 separate cilia from two fixed brain slices from one mouse). The starting point of tracing was chosen randomly. Based on the traced ciliary membrane, we counted the number of gaps, which we defined as a region with intensity smaller than a 2x standard deviation below the mean along the pGk13a labeled ciliary membrane, that was longer than 60 nm (in biological units, the effective resolution of the 60x 1.27NA objective that was used for imaging; Fig. 2g).

## umExM with double gelation (for corpus callosum)

For umExM with double gelation (for corpus callosum) (Supplementary Fig. 25), samples were prepared as described in the **umExM for brain tissue slices** section, except for fluorescently labeling the membrane probe and expansion (step 5-6). Then, the sample was incubated in a non-cleavable gelling solution (Supplementary Table 2, "Monomer solution") for 30 minutes on the shaker (~10-20 rpm) at 4 °C for pre-gelation incubation. We then gelled the sample at 37 °C, using the gelation chamber we described in step 3 of the **umExM for brain tissue slices** section. After the gelation, the initial gel was treated with a cleaving solution (50mM sodium metaperiodate in 0.1M sodium acetate buffer, pH 5.0) for one hour on the shaker (~100-150 rpm), at RT. Then the sample was washed 4 times in 100mM glycine PBS on the shaker (~50-100 rpm) at RT, 30 minutes each, and then the sample was washed 3–4 times with PBS on the shaker (~50–100 rpm) at RT, 15 minutes each. We then fluorescently labeled the membrane probe and expanded the sample as described in steps 5-6 of the **umExM for brain tissue slices** section.

## Accuracy (Rand score) of segmentation and tracing of pGk13a signals

We performed umExM with fixed brain slices from Thy1-YFP mice and boosted YFP signals with anti-GFP (as described in the **Antibody staining of fluorescent proteins for umExM** section). We imaged volumes of a random region in somatosensory cortex L6 and hippocampus dentate gyrus, with two labels (anti-GFP antibody and pGk13a for membranes).

**Segmentation.** To identify neuronal compartments, we generated a maximum-intensity z-projected (max-z projected) image from the anti-GFP channel of the volume. Using this max-z projected image, we pinpointed cell bodies, dendrites, and axons. However, anti-GFP signal alone cannot differentiate between myelinated and unmyelinated axons. We thus used the pGk13a signal to assist in identifying myelinated axons, as myelinated axons exhibited strong pGk13a signals compared to unmyelinated axons (Fig. 3r **for unmyelinated axon;** Fig. 3h **for myelinated axon**).

Subsequently, we randomly created several regions of interest (ROIs), each containing a portion of identified neuronal compartments. These ROIs were employed to crop the pGk13a channel and anti-GFP channel of the volume. We randomly selected a single z-plane from the cropped volume, manually segmented compartments based on pGk13a signals, and then segmented the same compartments based on anti-GFP signals, all with ITK-SNAP software[47]. We then

quantitatively compared the pGk13a-guided segmentation to the anti-GFP-guided segmentation using the Rand score[48,49]. We repeated this experiment and analysis ($n = 3$ cell bodies and $n = 3$ dendrites from two fixed brain slices from two mice, and $n = 5$ myelinated axons and $n = 5$ unmyelinated axons from two fixed brain slices from two mice).

### Tracing

We identified myelinated axons by inspecting anti-GFP signals as well as pGk13a signals in the same way as we described above. Among the identified myelinated axons, we randomly selected some. We traced them from $z = 0$ to $z = 10.5$ μm based on the pGk13a signals and also traced the same myelinated axons based on anti-GFP signals, all with ITK-SNAP software[47]. Specifically, we traced myelinated axons by annotating the centroid of the myelinated axons with brush size=8 in ITK-SNAP software. We then quantitatively compared the pGk13a-guided tracing to the anti-GFP-guided tracing using the Rand score. We repeated this experiment and analysis ($n = 3$ myelinated axons from two fixed brain slices from two mice). Next, we identified the unmyelinated axons by inspecting anti-GFP signals, as we did for the segmentation study above. We then randomly selected one and traced it from $z = 0$ to $z = 5$ μm based on the pGk13a and anti-GFP signals and calculated the Rand score[48,49], as we did for myelinated axons. We also repeated this experiment and analysis ($n = 3$ myelinated axons from two fixed brain slices from two mice).

For tracing myelinated axons in the corpus callosum, we applied umExM with double gelation protocol (Supplementary Fig 25) to mouse brain tissue section. We then imaged a random volume (39.25 by 39.25 by 20 μm) of the corpus callosum. We then used webKnossos[66] to trace $n = 20$ myelinated axons that spanned the entire dataset.

### umExM with Optical fluctuation imaging (umExM with SACD)

The samples were prepared as described in **umExM for brain tissue slices**. We imaged the samples with Andor spinning disk (CSU-W1 Tokogawa) confocal system with a 60x, 1.27NA objective with either a Zyla 5.5 camera or a Hamamatsu qCMOS, with an optional ×1.5 magnification. We used 20 frames of images (exposure time, 50ms; laser power 90%), which took ~1 second in total. We then used the SACD ImageJ plugin as provided in the Github (https://github.com/WeisongZhao/SACDj). We used the plugin with the default hyper-parameters[53] (i.e., 1st = 10, fourier=2, 2nd = 10, order=2, scale=2). Finally, CLAHE was applied for visualization purposes.

### ECS preservation protocol

ECS perfusion was adapted from the published protocol[56]. The mouse was terminally anesthetized with isoflurane and placed on a dissection tray. The chest was cut open, and a 21-gauge butterfly needle was inserted into the left ventricle. A small incision was made in the right atrium to facilitate outflow. The mouse was perfused transcardially at a flow rate of 10 mL/min using a Masterflex Peristaltic pump. Fresh aCSF was flown for 2-3 minutes to clear out the blood. This was followed by perfusion with 15% mannitol in aCSF solution for 1 minute, and then a 6% mannitol aCSF solution for 5 minutes. Finally, the mouse was perfused with an ice-cold fixative containing 5% mannitol, 4% paraformaldehyde, 2mM $CaCl_2$, 4mM $MgCl_2$, and 150mM sodium cacodylate buffer (pH 7.4) for 5 minutes.

After perfusion, the brain was carefully removed from the skull and placed in a vial containing the same fixative solution. It was then fixed for at least 24 hours with gentle agitation at 4 °C. 100μm sections were cut using a Leica VT1000 S vibrating blade microtome and collected in the cold fixative solution.

### Iterative form of umExM

For the iterative form of umExM (Supplementary Fig. 26), umExM samples were prepared as described in the **umExM for brain tissue slices** section, except for fluorescently labeling the membrane probe

(step 5). The expanded samples were incubated in a cleavable re-embedding solution (Supplementary Table 3, "Second gelling solution") for 1 hour on a shaker (~50 rpm) at RT for pre-gelation incubation. Next, we gelled the sample at 50 °C, for >4 hours, with the same gelation chambers used in step 3 of the **umExM for brain tissue slices** section. The re-embedded samples were washed 3–4 times in PBS at RT, 30 minutes each. The re-embedded samples were then treated with AX solution and washed in PBS as described in step 2 of the **umExM for brain tissue slices** section. The samples were trimmed into smaller samples with razor blades and then gelled again with a non-cleavable gelling solution (Supplementary Table 3, "Third gelling solution") 30 minutes on the shaker (~10 – 20 rpm) at 4 °C for pre-gelation incubation. Next, we gelled the sample at 37 °C, overnight, with the same gelation chambers used above. The samples were treated with the cleaving solution (50mM sodium metaperiodate in 0.1M sodium acetate buffer, pH 5.0) for one hour, at RT. Then the samples were washed 4 times in 100mM glycine PBS on the shaker (~50–100 rpm) at RT, 30 minutes each, and then the sample was washed 3-4 times with PBS on the shaker (~50–100 rpm) at RT, 15 minutes each. We then fluorescently labeled the membrane probe and expanded the sample as described in steps 5-6 of the **umExM for brain tissue slices** section.

### Statistics & reproducibility

No statistical method was used to predetermine sample size. No data were excluded from the analyses.

### Reporting summary

Further information on research design is available in the Nature Portfolio Reporting Summary linked to this article.

## Data availability

The raw and processed image stack data generated with umExM in this study are available on the Open Science Framework at https://osf.io/qtbek/. Source data are provided with this paper.

## Code availability

The source code for analyzing umExM data is available on GitHub at https://github.com/TAYmit/umExM

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

## Acknowledgements

We thank Sebastian Seung and Jeremy Baskin for their thoughtful discussions. We thank Chih-Chieh (Jay) Yu for providing training, and to Yixi Liu and Amy Yang for their assistance with cell preparation. We also thank Zeguan Wang for assisting with the measurement of laser excitation power. We appreciate the valuable comments from Burcu Guner-Ataman. We thank Camille Mitchell and Daniel Lieble for their help with antibody staining. We thank the valuable discussion with Anu Sinha and Brett Pryor. Figure 1, Supplementary Figs. 20-21, 23, 25 and 26 were created using Biorender.com. For funding, E.S.B. acknowledges Lisa Yang and Y. Eva Tan, John Doerr, the Open Philanthropy Project, the MIT Media Lab, HHMI, the US Army Research Laboratory and the US Army Research Office under contract/grant number W911NF1510548, Cancer Research UK Grand Challenge grant C9545/A24042, the MIT Brain and Cognitive Sciences Department, the New York Stem Cell Foundation-Robertson Investigator Award, NIH Transformative Award 1R01GM104948, NIH Director's Pioneer Award 1DP1NS087724, NIH 1R01EY023173, NIH 1U01MH106011, the MIT McGovern Institute MINT program, Lore McGovern, Tom Stocky and Avni Shah, Kathleen Octavio, Good Ventures, NIH 1R01AG070831, NIH 1R01MH123403, NIH R01MH124606, NIH R01AG087374 and Schmidt Futures. T.W.S was supported by NSF Graduate Research Fellowship grants 1122374 and J. Douglas Tan Postdoctoral Fellow for Autism Research. J.S.K. acknowledges funding from the Samsung Scholarship.

## Author contributions

T.W.S. and E.S.B. spearheaded the study, incorporating insights from L.-H.T. and receiving early input from E.D.K., J.S.K., and A.H.M. Designing and optimizing the probe, as well as developing the relevant ExM protocol, was done by T.W.S. with help from C.Z. and early input from E.D.K. and L.S.K. Experimental work was carried out by T.W.S., H.W., C.Z., B.A., Y.L., E.Z., X.L., E.D.K., J.S.K., L.L., E.K.C., A.E., and N.K. Data analysis was performed by T.W.S., H.W., C.Z., E.D.K and P.S. Mouse perfusion, fixation, and tissue preparation were handled and performed by T.W.S., B.A., J.S.K., A.E., L.L., and E.K.C. Imaging was performed by T.W.S., H.W., J.S.K., A.E., L.L., and E.K.C. T.W.S. wrote the manuscript, incorporating valuable insights and/or edits from C.Z., L.-H.T., and E.S.B. The project was supervised by E.S.B.

## Competing interests

T.W.S. and E.S.B. are co-inventors on a patent application for umExM (No.: 63/520,702). E.S.B. is co-founder of a company seeking to deploy applications of ExM-related technologies. The other authors declare no competing interests.

## Additional information

## IMAXT Grand Challenge Consortium

**Edward S. Boyden** ⓘ [1,2,3,6,9,10,11,12] ✉

