## [Transparent Peer Review file · Nature Communications]

Dense, Continuous Membrane Labeling and Expansion Microscopy Visualization of Ultrastructure in Tissues

Corresponding Author: Professor Edward Boyden

Version 0:

Reviewer comments:

Reviewer #1

(Remarks to the Author)

The article by Shin and colleagues focuses on the development of a novel molecule designed for visualizing expanding lipids. In this study, they conjugated a lipid with multiple lysines to facilitate its anchoring in the expanding polymer and utilized an azide as a chemical handle. The authors convincingly demonstrate the functionality of this molecule in cells and tissues, employing both optical and photonic microscopy techniques. Furthermore, they exhibit a resolution of approximately 60nm using FRC and highlight the compatibility of this approach with other expansion microscopy labeling methods such as immunolabeling or FISH labeling. Lastly, the authors showcase the feasibility of employing automatic segmentation techniques with this labeling strategy and its potential coupling with other super-resolution optical fluctuation techniques.

While I found the article interesting and recognize the amount of work that has gone into demonstrating the effectiveness of the probe, I find that the article suffers from the following major shortcomings:

- There are now several articles demonstrating the use of probes to mark lipids (non-exhaustive list below), and it seems that they are not all cited. More importantly, there is no comparison with known probes such as mCLING or BODIPY. For example, BODIPY is added post-expansion, and many groups have observed that lipids are retained, depending on the fixation. A comparison with existing probes is therefore needed to demonstrate that this new molecule is better and that it is the only one to provide dense, continuous labeling:

Biotin-DHPE with Tyramide Signal Amplification (TSA) :
<https://www.nature.com/articles/s41598-023-48959-9>

Sphingolipid ExM, both sides of the mitochondrion are visible
<https://www.nature.com/articles/s41467-020-19897-1>

mCLING in human cells and Caco2 monolayer apical brush border
<https://elifesciences.org/articles/73775> (paper cited but not for lipid ExM)

SARS-CoV-2-infected ciliated cells
<https://www.biorxiv.org/content/10.1101/2021.08.05.455126v1>

As well as BODIPY application in post-expansion labeling:

BODIPY:
Chlamydomonas
<https://www.ncbi.nlm.nih.gov/pmc/articles/PMC10502176/>

Plasmodium:
<https://www.mdpi.com/2076-2607/9/11/2306>

human cells:
<https://journals.plos.org/plosone/article?id=10.1371/journal.pone.0291506>

- The resolution calculated here is not the effective resolution but rather the FRC, which, when used with a confocal or a widefield, determines the system resolution, i.e., 200-240nm. Subsequently, as the sample is 4 times larger, the resolution value is divided by the expansion factor, resulting in approximately 60nm. However, I find that this calculation may not accurately reflect reality. Could a demonstration of the effective resolution be performed by measuring microtubules or even the cristae of mitochondria, which should be discernible with a 60nm resolution?

- An indication that the resolution is not 60nm is presented in Figure S2C, where greater resolution is observed in SIM (panel c) than after expansion (d). I acknowledge the difference in staining, but both sides of the mitochondria should be visible after expansion if the resolution is indeed 60nm. Similarly, for Figure 2j, an inset on the structure showing a 200nm structure would be great to demonstrate the effective resolution.

- One of my primary concerns is that the probe should not mark all membranes but should be selective for specific regions or organelles. In figures 2a-c, S2a, or S3a, mainly mitochondria are observed, while cytoplasmic vesicles, reticulum, or other lipid-based organelles are not. This result is even more apparent in S1, where vesicle staining is absent, as are mitochondria cristae. While this outcome may be normal due to variations in lipid composition within the plasma membrane and among organelles, it suggests that the probe may only intercalate with a particular type of membrane. However, this information is not mentioned in the article or compared with existing probes.

- Additionally, the authors introduce an optimization to retain more lipids during gelation, calling it ultrastructural mExM. I find the term "ultrastructural" to be inaccurate in this context, as 4% PFA fixation does not properly preserve ultrastructure, making the term ultrastructure misleading. The authors may note that lipids are less extracted during fixation, but they should refrain from using the term "ultrastructural." In electron microscopy, the preservation of ultrastructure typically requires a strong glutaraldehyde-based fixation.

Reviewer #2

(Remarks to the Author)

Expansion microscopy (ExM) has transformed the way we examine the spatial arrangements of biomolecules - mostly proteins - across various types of samples. This study introduces a new ExM technique - the author termed "umExM", or "ultrastructural membrane expansion microscopy" - that further expands the ExM toolbox by providing a way to better visualize membranes, especially in dense tissue samples. The substrate used is newly designed and synthesized and clearly thought out, while the resulting data shows EM-like labeling, especially after image deconvolution. In general, the conclusions are well supported by the data. The method in the current state is readily useful for many applications while providing a solid foundation for further optimizations.

My comments are below. The comment in which additional experiments would be needed is regarding Line 181 (3).

Line 76: can the authors comment on the ideal slice thickness, for example, could 50 or 200 micron thick samples be used?

Line 78: high signal to noise ratio - would be helpful to provide a number (40 to 80 signal versus background, as described in the main text)

Line 128 and supplementary Fig 1: the main text talked about pGk5b. However, the images here were from pGk5a stainings. Might need to move the figure to the later part of the paper.

Line 134, and supplementary Fig 2c-2l: would be helpful to add an annotation in the figures showing that panels c-f are from GFP, and g-i are from pGk5b staining. It seems that, in these comparisons, the membrane labeling images are collected only post-expansion, but not pre-expansion (see also Fig 2). It would be helpful for the authors to recognize this point.

Line 138, and supplementary Fig 3c: the bars are not very helpful for visualizing the data. Something similar to supplementary Fig. 2i (box graph) would be better and provides a consistent styling throughout the paper.

Line 140, and supplementary Fig. 4: there is a huge increase in contrast post-expansion. Can the authors provide possible explanations?

Line 145, and supplementary Fig 5: again, the bars in (d) obscures the data points.

Line 154, and supplementary Fig 6: have the authors tested both farnesylated and palmitoylated probes applied to the same sample? Presumably, they might attach to lipid bilayers with different properties and thus provide a signal boost compared to using either one alone.

Line 162, and supplementary Fig 8: the mitochondria are stained intensely. Some of these signals could be from endogenously biotinylated proteins that are commonly found on mitochondria (<https://doi.org/10.1385/1-59745-579-2:111>). This can be a good point to mention when the authors replaced biotin with azide (Line 179).

Line 171, and supplementary Fig. 9: perhaps a better mouse line to use is membrane-EGFP mouse line (<https://www.jax.org/strain/007676>). However, this is relatively a minor point.

Line 181, and supplementary Fig 10:

(1) Again, the bars in (c) are not very helpful.

(2) Have the authors explored other handles, such as small peptide tags? This can perhaps be in the discussion section for future developments.

(3) DBCO itself is lipophilic and can contribute to the membrane signal. It would be helpful to conduct a control experiment in which only DBCO-dye is used.

Line 197 and supplementary Fig 11: the bars should be avoided in these graphs (11c).

Line 236-239, and Figure 2q-r: these are motile cilia (about a dozen per cell on ependymal cells and airway epithelium, etc) and not primary cilia (one per cell, found on neurons, beta cells in the pancreas, among many others). The text in 2r would need to be changed. In addition, the references are for primary cilia (38-30) and need to be replaced (for an example, <https://doi.org/10.3109/2000-1967-010>). n-r and v: the bars are not helpful for visualization.

Line 243, and supplementary Fig 13: choroid plexus cells also have primary cilia

(<https://doi.org/10.1016/j.devcel.2023.10.003>). Just curious - did the authors see these in their datasets?

Line 254: throughout the text, the authors used laser exposure time as an indication of the amount of the laser excitation/light dose used. Without knowing the laser excitation power (mW), however, these numbers lack context. Would be helpful to establish a baseline by standard samples, for example, a cell expresses regular cytoplasmic EGFP.

Line 257 and the remainder of the paragraph: related to the statement above - was the same laser power used?

Line 275, and Fig. 2t-v: it would be helpful to point out that these are motile cilia (see comment #13 above).

Line 282, and Fig 2v: can the authors comment a bit more in regards to the context of a gap of 60 nm?

Line 350: Fig 4e should be mentioned here in the text. The bars in the graph are not that helpful (see also comments above).

Line 395: should this modified protocol be used in other areas of the brain as well? A brief mention of this would suffice.

Reviewer #3

(Remarks to the Author)

The manuscript introduces ultrastructural membrane expansion microscopy (umExM), a novel technique for high-resolution imaging of cellular membranes. The authors achieve high label density through a multi-faceted approach centered on the development and application of novel synthetic amphiphilic membrane labeling probes. These probes are designed to densely and uniformly label membrane structures, including plasma membranes and various organelle membranes, within fixed tissues. The core ideas behind achieving such high label density include:

- The membrane labeling probes exhibit lipophilicity, akin to traditional fluorescent lipophilic dyes. This characteristic enables the probes to preferentially localize and diffuse within membrane structures, ensuring comprehensive membrane coverage.
- The probes are designed with a chemical handle that allows for the selective attachment of fluorophores after the formation of the ExM polymer network. This feature ensures that the labeling probes remain small, facilitating their diffusion within the tissue and preserving the fluorophore's integrity during the polymerization process.
- The probes possess a polymer-anchorable handle, enabling their incorporation into the swellable polymer network created during the ExM process. This integration is crucial for the physical expansion of the labeled membranes alongside the hydrogel, maintaining the high density of labeling through the expansion process.
- Through systematic design and optimization, including the selection of the hydrocarbon side chains and the number of lysines in the probe backbone, the authors tailored the probes for optimal performance. This optimization process involved exploring the chemical space to identify the probe configurations that best support dense membrane staining with minimal distortion and high fidelity.
- The ExM protocol itself is optimized to accommodate these novel probes, including adjustments to the chemical softening of the sample, the temperature control during processing, and the specific conditions for hydrogel formation and expansion. These adjustments ensure the preservation of membrane integrity and the effectiveness of labeling throughout the ExM process.

The authors apply their novel methodology to fixed mouse brain tissue. They specifically focus on labeling and visualizing various membranous structures within this tissue, including:

- Plasma Membranes: The outer membranes of cells that act as the boundary between the cell interior and the external environment.
- Mitochondrial Membranes: The membranes enclosing mitochondria, the cell's powerhouse, involved in energy production.
- Nuclear Membranes: The double membrane structure surrounding the nucleus, where genetic material is stored.
- Ciliary Membranes: The membrane of cilia, which are small hair-like structures that protrude from the cell body and are involved in movement and sensing the external environment.
- Myelin Sheaths: The fatty membrane that ensheathes and insulates axons, enhancing the speed of electrical signal transmission in the nervous system.
- Extracellular Vesicle Membranes: The membranes surrounding extracellular vesicles, which are small particles released

from cells that can carry signaling molecules to other cells.

By applying umExM to these diverse membrane systems, the authors are able to achieve dense and continuous labeling with high signal-to-background ratios, enabling the visualization of membranous structures at a resolution of approximately 60 nm. This methodology supports co-visualization of membranous structures alongside proteins and RNAs, allowing for the segmentation of neuronal compartments such as cell bodies, dendrites, and axons, and tracing of neuronal processes.

The manuscript is well-structured and logically coherent, effectively conveying the significance and innovation of umExM. I could not find any linguistic or logical errors. Therefore, I recommend publication of the manuscript as is.

Version 1:

Reviewer comments:

Reviewer #1

(Remarks to the Author)

I would like to thank the authors for their work and for answering my questions so thoroughly. Overall this manuscript represents a high-quality contribution to the field, with an impressive range of controls and analyses that strengthen the validity of the results. I have no further comments to make and fully support its publication.

Reviewer #2

(Remarks to the Author)

The authors had done a great job responding to my previous comments in the revised manuscript. I only have one very minor comment:

In Supplementary Figure 12 legend (Line 433-435):

a) Representative (n=2 fixed brain slices from two mice) single z-plane confocal image of expanded mouse brain tissue (CA1), that underwent the umExM protocol, but without pGk13a staining of the membrane, showing DBCO-Cy3 staining of the membrane (inverted gray).

Since in non-pGk12b conditions, barely any signal was present. Therefore, I believe a better description would be:

a) Representative (n=2 fixed brain slices from two mice) single z-plane confocal image of expanded mouse brain tissue (CA1), that underwent the umExM protocol, but without pGk13a staining of the membrane, showed negligible signals (inverted gray).

The manuscript is otherwise beautifully written with wonderful results, and will be very helpful to advance of the field in ExM and thus potentially uncover new biology.

We have now responded to all the reviewers' comments with additional work, and we hope that our paper is now acceptable at Nature Communications.

Reviewer #1 (Remarks to the Author):

The article by Shin and colleagues focuses on the development of a novel molecule designed for visualizing expanding lipids. In this study, they conjugated a lipid with multiple lysines to facilitate its anchoring in the expanding polymer and utilized an azide as a chemical handle. The authors convincingly demonstrate the functionality of this molecule in cells and tissues, employing both optical and photonic microscopy techniques. Furthermore, they exhibit a resolution of approximately 60nm using FRC and highlight the compatibility of this approach with other expansion microscopy labeling methods such as immunolabeling or FISH labeling.

1. We thank the reviewer for their appreciative comments.

Lastly, the authors showcase the feasibility of employing automatic segmentation techniques with this labeling strategy and its potential coupling with other super-resolution optical fluctuation techniques.

2. Regarding "the authors showcase the feasibility of employing automatic segmentation techniques with this labeling strategy" - we note that we did not employ automatic segmentation techniques in this work. All tracing was done manually.

While I found the article interesting and recognize the amount of work that has gone into demonstrating the effectiveness of the probe, I find that the article suffers from the following major shortcomings:

- There are now several articles demonstrating the use of probes to mark lipids (non-exhaustive list below), and it seems that they are not all cited.

3. Regarding "There are now several articles demonstrating the use of probes to mark lipids (non-exhaustive list below), and it seems that they are not all cited", we had included citations for all membrane or lipid labeling ExM technologies that we were aware of in Supp. Table 1 or its caption (the table itself went on to summarize the properties of the subset of techniques that had been used in tissues), which included all the mentioned chemistries except for Biotin-DHPE ExM technology¹, which we were not aware of when submitting the paper. We have now added Biotin-DHPE¹ to the list. For each lipid stain used in an ExM context, we tried to choose one representative study as a citation (aiming to cite the first published study to use a given stain). For example, for mCling, we cited TReX² amongst TReX² and ref³. Finally, we've cited all of these citations in the main text, in the introduction.

More importantly, there is no comparison with known probes such as mCling or BODIPY. For example, BODIPY is added post-expansion, and many groups have observed that lipids are retained, depending on the fixation. A comparison with existing probes is therefore needed to demonstrate that this new molecule is better and that it is the only one to provide dense, continuous labeling:

Biotin-DHPE with Tyramide Signal Amplification (TSA) :
<https://www.nature.com/articles/s41598-023-48959-9>

Sphingolipid ExM, both sides of the mitochondrion are visible
<https://www.nature.com/articles/s41467-020-19897-1>

mCLING in human cells and Caco2 monolayer apical brush border
<https://elifesciences.org/articles/73775> (paper cited but not for lipid ExM)

SARS-CoV-2-infected ciliated cells
<https://www.biorxiv.org/content/10.1101/2021.08.05.455126v1>

As well as BODIPY application in post-expansion labeling:

BODIPY:
Chlamydomonas
<https://www.ncbi.nlm.nih.gov/pmc/articles/PMC10502176/>

Plasmodium:
<https://www.mdpi.com/2076-2607/9/11/2306>

human cells:
<https://journals.plos.org/plosone/article?id=10.1371/journal.pone.0291506>

4. Regarding “More importantly, there is no comparison with known probes such as mCling or BODIPY. For example, BODIPY is added post-expansion, and many groups have observed that lipids are retained, depending on the fixation. A comparison with existing probes is therefore needed to demonstrate that this new molecule is better and that it is the only one to provide dense, continuous labeling”, we tested BODIPY-lipid, mCling and Biotin-DHPE, among the suggested lipid stains, as these three were commercially available (i.e., the probe used in ref⁴, was not commercially available). Note that none of these lipid stains had been reported for tissue application in the ExM context, except mCling, which, in a study using it in tissue, did not provide much in the way of methodological detail⁵. Given the lack of tissue protocol available for these probes, we utilized the staining protocols that had been established for cultured cells. These protocols were applied to standard 4% PFA-fixed brain tissue, and we performed the most commonly used form of ExM, proExM⁶ (see Supplementary Methods for details).

Subsequently, we performed the same signal-to-background analysis and continuity analysis as we did for umExM, finding the following:

a. Signal-to-background (S/B) analysis:

We imaged a random part of the CA1 region of the mouse hippocampus with a 4x objective at 50ms laser exposure time for samples that were stained with biotin-DHPE, BODIPY FL C12, or mCling (**Supp Fig. 17a-c**). We also performed umExM (pGk13a) for comparison (**Supp Fig. 17d**). Qualitatively, umExM generated the highest contrast image, compared to the others. We then measured the signal-to-background (S/B, where the background was determined as the average across images of empty gel regions) for the images obtained from each sample. We found that the S/B for umExM images was many times higher than those of Biotin-DHPE, BODIPY FL C12, and mCling sample images. It is in principle possible that such stains could be chemically modified further, or their tissue protocol optimized to improve staining beyond what has been published to date, but these would be entire studies by themselves, beyond the scope of our work.

b. Continuity analysis:

We imaged cilia in the 3rd ventricle to perform continuity analysis, as we did in **Fig. 2v**. However, except for the sample stained with mCling, we were not able to observe any signals. Furthermore, although mCling was able to visualize cilia to some extent, the signals were not dense enough for membranes to be traced, so continuity analysis was not possible (**Supp Fig. 17f-g**).

We have added these results to the main text and the supplementary notes of our manuscript.

- The resolution calculated here is not the effective resolution but rather the FRC, which, when used with a confocal or a widefield, determines the system resolution, i.e., 200-240nm. Subsequently, as the sample is 4 times larger, the resolution value is divided by the expansion factor, resulting in approximately 60nm. However, I find that this calculation may not accurately reflect reality. Could a demonstration of the effective resolution be performed by measuring microtubules or even the cristae of mitochondria, which should be discernible with a 60nm resolution?

5. Regarding “the resolution calculated here is not the effective resolution but rather the FRC, which, when used with a confocal or a widefield, determines the system resolution, i.e., 200-240nm. Subsequently, as the sample is 4 times larger, the resolution value is divided by the expansion factor, resulting in approximately 60nm. However, I find that this calculation may not accurately reflect reality.”, we respectfully need to correct the reviewer’s comment. FRC analysis does not know anything about the imaging system, but instead, simply quantifies the effective resolution of the provided images. FRC analysis has been widely used on images to measure the effective resolution of an entire

imaging process, not just the microscope itself^{7,8}. Since we performed FRC analysis on images of expanded (i.e., umExM) samples, FRC would be assessing the resolution of those very images; the FRC algorithm is not provided with the magnification of our system, nor the expansion factor, and thus cannot shed any light on the system resolution. Note well – in order to provide resolution in biological units, we did normalize the pixel size of our umExM images by the expansion factor, so that the resolution would be described in biologically relevant terms – perhaps this is the source of the confusion here. (Just to make this clear, we added a sentence to the methods section, explicitly mentioning that we normalized pixel size by expansion factor.) In short, we performed umExM, imaged samples, corrected pixel size to biological units, and then performed FRC analysis. This allowed us to measure the effective resolution of umExM images themselves, instead of the system resolution of the microscope.

6. Regarding “Could a demonstration of the effective resolution be performed by measuring microtubules”: we considered this, but established methods for microtubule staining for measuring resolution (which our group, and others, have done in the past for ExM validation) requires a specific protocol (i.e., cytoskeleton extraction, tubulin fixation (PFA + glutaraldehyde fixation), and antibody staining)^{2,8,9}, to get clean microtubule images. This process requires a high concentration of detergent (e.g., 0.2% triton) that removes membrane lipids, which would compromise downstream umExM processing.
7. Regarding “or even the cristae of mitochondria, which should be discernible with a 60nm resolution?”: it requires ~30nm resolution to resolve cristae¹⁰. Therefore, mitochondria cristae cannot be resolved with the 4x expansion factor form of umExM. However, the iterative form of umExM shows mitochondrial cristae (**Supp. Fig. 25**), showing similar appearance as seen with isoSTED (~30nm resolution imaging; Fig. 2b from ref¹⁰)

- An indication that the resolution is not 60nm is presented in Figure S2C, where greater resolution is observed in SIM (panel c) than after expansion (d). I acknowledge the difference in staining, but both sides of the mitochondria should be visible after expansion if the resolution is indeed 60nm. Similarly, for Figure 2j, an inset on the structure showing a 200nm structure would be great to demonstrate the effective resolution.

8. Regarding “An indication that the resolution is not 60nm is presented in Figure S2C, where greater resolution is observed in SIM (panel c) than after expansion (d). I acknowledge the difference in staining, but both sides of the mitochondria should be visible after expansion if the resolution is indeed 60nm”, we expressed mitochondrial matrix-targeted (e.g., non-membrane targeted) GFP via BacMam virus, since the goal of this experiment was to gauge distortion of the sample. Thus, **Supp. Fig. 2D** cannot visualize the membrane of mitochondria.

9. Regarding “Similarly, for Figure 2j, an inset on the structure showing a 200nm structure would be great to demonstrate the effective resolution.”, we show several images that visualize structures at this length scale or smaller. We provided a single z-plane umExM image showing cilia (**Fig. 2t**) as well as serial sections showing cilia (**Supp. Fig. 13**), a z-projection of cilia (**Fig. 2q**), and a volume rendering of cilia (**Fig. 2s**). As umExM provides ~60nm effective resolution (see Authors’ response 5), ~200nm structures are clearly resolved and visualized.

- One of my primary concerns is that the probe should not mark all membranes but should be selective for specific regions or organelles. In figures 2a-c, S2a, or S3a, mainly mitochondria are observed, while cytoplasmic vesicles, reticulum, or other lipid-based organelles are not. This result is even more apparent in S1, where vesicle staining is absent, as are mitochondria cristae. While this outcome may be normal due to variations in lipid composition within the plasma membrane and among organelles, it suggests that the probe may only intercalate with a particular type of membrane. However, this information is not mentioned in the article or compared with existing probes.

10. Regarding “One of my primary concerns is that the probe should not mark all membranes but should be selective for specific regions or organelles. In figures 2a-c, S2a, or S3a, mainly mitochondria are observed, while cytoplasmic vesicles, reticulum, or other lipid-based organelles are not”, this may be due to the limited resolution of these images, rather than due to variations in membrane labeling. Indeed, electron microscopy (EM)-processed samples with low resolution imaging cannot visualize small organelles either. For example, X-ray imaging of samples processed using classical EM techniques (i.e., PFA + glutaraldehyde fixation followed by osmium staining and uranyl acetate staining), offers ~83nm resolution imaging¹¹. However, many organelles, except for mitochondria (see image below; data from ref¹¹) and endoplasmic reticulum (ER, Fig 1g inset from ref¹¹), were not visualized, due to the low resolution:

Similar to this EM sample processing + X-ray imaging outcome, our protocol visualizes both mitochondria (**Supp. Fig. 2a,b,g; Supp. Fig. 3a**) and ER (**Supp. Fig. 3b; Fig.**

S4aiv; Supp. Fig. 7a), and can reveal some features of cytoplasmic vesicles (i.e., synaptic vesicles; Fig. 3 c,d,e,f), although the round shapes of these vesicles cannot be seen with the resolution of 4x umExM. Thus, it is possible that what is seen in such images, is simply due to the poor resolution of 4x umExM, compared to EM.

With the iterative form of umExM, which provides higher resolution (i.e., ~35nm resolution, **Fig. 6g**) compared to umExM, we observed mitochondria cristae (**Supp. Fig. 25**) showing a similar topology as what was seen with isoSTED¹⁰. We also saw ER-like structures, but we again did not see the round shapes of synaptic vesicles, perhaps due to limited stain density in conjunction with borderline resolution. In umExM images obtained with SACD, we did not observe mitochondrial cristae, despite the higher resolution; it may be necessary to optimize SACD parameters⁷ to see this. We have commented on this, in the discussion section.

11. Regarding “This result is even more apparent in S1, where vesicle staining is absent, as are mitochondria cristae.”, we note that S1 showed EM images. EM sample processing, involving PFA+glutaraldehyde fixation followed by osmium staining, has been optimized over decades¹²⁻¹⁴, and the clear visualization of organelles is protocol-dependent. Early EM protocols often yielded images where mitochondrial cristae and synaptic vesicles were challenging to identify (e.g., see Fig 6,9,10 from ref¹⁵), but further optimization eventually helped. Similarly, further optimization of EM processing for the context of our specific membrane probe could lead, in principle, to clear visualization of vesicles and mitochondrial cristae through our lipid stain used in conjunction with EM imaging. However, since our objective was not to optimize our membrane probe treatment and sample processing for EM imaging, but instead to produce EM-like images with expansion microscopy and confocal imaging, we did not further pursue protocol optimization for EM imaging. This explanation has been added to **Supplementary Note 1**.
12. Regarding “While this outcome may be normal due to variations in lipid composition within the plasma membrane and among organelles, it suggests that the probe may only intercalate with a particular type of membrane.”, we agree that this may be possible, although other possibilities may be occurring here too (Authors’ response 10, Authors’ response 11). We add a note acknowledging these possibilities to the Discussion.

- Additionally, the authors introduce an optimization to retain more lipids during gelation, calling it ultrastructural mExM. I find the term "ultrastructural" to be inaccurate in this context, as 4% PFA fixation does not properly preserve ultrastructure, making the term ultrastructure misleading. The authors may note that lipids are less extracted during fixation, but they should refrain from using the term "ultrastructural." In electron microscopy, the preservation of ultrastructure typically requires a strong glutaraldehyde-based fixation.

13. Regarding “Additionally, the authors introduce an optimization to retain more lipids during gelation, calling it ultrastructural mExM. I find the term "ultrastructural" to be inaccurate in this context, as 4% PFA fixation does not properly preserve ultrastructure, making the term ultrastructure misleading. The authors may note that lipids are less extracted during fixation, but they should refrain from using the term "ultrastructural.” - we were using the word "ultrastructure" not because we retain more lipids during gelation, but instead we were using the word in the same sense as an earlier protocol in the expansion microscopy community, called ultrastructure expansion microscopy (u-ExM¹⁶). We note that this earlier paper did not claim to retain any lipids at all, but still used the word ultrastructure. The definition of ultrastructure is “biological structure and especially fine structure (as of a cell) not visible through an ordinary microscope” (Merriam-Webster), and we are simply trying to use the word as it has been used in the expansion microscopy community before. We now point out that we are using the word in the expansion microscopy community sense, in the revised paper. In the end, of course, we would defer to the editor as to this point.

14. Regarding “In electron microscopy, the preservation of ultrastructure typically requires a strong glutaraldehyde-based fixation”, we agree that EM typically requires strong glutaraldehyde-based fixation, but there are many studies that use low-glutaraldehyde preservation solutions for EM^{17,18}. Strong glutaraldehyde-based fixation serves to protect the sample during the steps of EM staining and electron imaging^{12,14}, but may not be needed here. Indeed, for expansion to work, molecules must be separable from each other, meaning that they cannot be over-crosslinked to each other, the way glutaraldehyde would make things. Our goal is not to preserve ultrastructure for EM processing and imaging, but rather to preserve it through ExM processing and confocal imaging.

Reviewer #2 (Remarks to the Author):

Expansion microscopy (ExM) has transformed the way we examine the spatial arrangements of biomolecules - mostly proteins - across various types of samples. This study introduces a new ExM technique - the author termed "umExM", or "ultrastructural membrane expansion microscopy" - that further expands the ExM toolbox by providing a way to better visualize membranes, especially in dense tissue samples. The substrate used is newly designed and synthesized and clearly thought out, while the resulting data shows EM-like labeling, especially after image deconvolution. In general, the conclusions are well supported by the data. The method in the current state is readily useful for many applications while providing a solid foundation for further optimizations.

My comments are below. The comment in which additional experiments would be needed is regarding Line 181 (3).

15. We thank the reviewer for the complimentary, thoughtful, and constructive comments.

Line 76: can the authors comment on the ideal slice thickness, for example, could 50 or 200 micron thick samples be used?

16. Regarding “can the authors comment on the ideal slice thickness, for example, could 50 or 200 micron thick samples be used?”, we tested our protocol on slices from 50 to 100 microns thick. We did not test the procedure on tissue slices thicker than 100 microns, because our current focus was on the chemistry of lipid staining, but we see no reason why this wouldn't work. Expansion microscopy protocols have been extensible to very large samples, even entire mouse brains¹⁹. Thicker samples may require longer pGk13a incubation times, or a higher concentration of pGk13a, or both, as slice thickness increases.

Line 78: high signal to noise ratio - would be helpful to provide a number (40 to 80 signal versus background, as described in the main text)

17. Regarding “high signal to noise ratio - would be helpful to provide a number (40 to 80 signal versus background, as described in the main text)”, we have provided a number as the reviewer suggested.

Line 128 and supplementary Fig 1: the main text talked about pGk5b. However, the images here were from pGk5a stainings. Might need to move the figure to the later part of the paper.

18. Regarding “Line 128 and supplementary Fig 1: the main text talked about pGk5b. However, the images here were from pGk5a stainings. Might need to move the figure to the later part of the paper.”: pGk5a and pGk5b are very similar; the only difference is in the group that can be later modified to attach a visualizable tag. Specifically, we used pGk5a to bind gold nanoparticles via DBCO for EM imaging, whereas pGk5b was used to bind fluorescent streptavidin for light microscopy imaging. The use of pGk5b in the main text was a typo – we have changed this to pGk5a.

Line 134, and supplementary Fig 2c-2l: would be helpful to add an annotation in the figures showing that panels c-f are from GFP, and g-i are from pGk5b staining. It seems that, in these comparisons, the membrane labeling images are collected only post-expansion, but not pre-expansion (see also Fig 2). It would be helpful for the authors to recognize this point.

19. Regarding “Line 134, and supplementary Fig 2c-2l: would be helpful to add an annotation in the figures showing that panels c-f are from GFP, and g-i are from pGk5b staining. It seems that, in these comparisons, the membrane labeling images are collected only post-expansion, but not pre-expansion (see also Fig 2). It would be helpful for the authors to recognize this point.”, we have added such annotations to **Supp. Fig. 2c-e,g-h** and **Fig. 2a-c**.

Line 138, and supplementary Fig 3c: the bars are not very helpful for visualizing the data. Something similar to supplementary Fig. 2i (box graph) would be better and provides a consistent styling throughout the paper.

20. Regarding “Line 138, and supplementary Fig 3c: the bars are not very helpful for visualizing the data. Something similar to supplementary Fig. 2i (box graph) would be better and provides a consistent styling throughout the paper.”, we changed **Supp. Fig. 3c** to a box graph as suggested by the reviewer. Furthermore, we changed all bar graphs to boxplots to have consistent styling throughout the paper, as the reviewer suggested.

Line 140, and supplementary Fig. 4: there is a huge increase in contrast post-expansion. Can the authors provide possible explanations?

21. Regarding “Line 140, and supplementary Fig. 4: there is a huge increase in contrast post-expansion. Can the authors provide possible explanations?”, the pre-expansion images of mouse brain tissue in **Supp. Fig. 4** contain native lipids, which were not removed before imaging. Accordingly, the pre-expansion sample exhibits substantial light scattering and a mismatch in refractive index, which significantly impacts image contrast²⁰. Meanwhile, both light scattering as well as mismatch in refractive index problems are ameliorated after ExM²¹. As a result, there is a huge increase in contrast post-expansion. This explanation has been added to the legend of **Supp. Fig 4**.

Line 145, and supplementary Fig 5: again, the bars in (d) obscures the data points.

22. Regarding “Line 145, and supplementary Fig 5: again, the bars in (d) obscures the data points..”, we changed **Supp. Fig. 5d** to a box graph as suggested by the reviewer.

Line 154, and supplementary Fig 6: have the authors tested both farnesylated and palmitoylated probes applied to the same sample? Presumably, they might attach to lipid bilayers with different properties and thus provide a signal boost compared to using either one alone.

23. Regarding “Line 154, and supplementary Fig 6: have the authors tested both farnesylated and palmitoylated probes applied to the same sample? Presumably, they might attach to lipid bilayers with different properties and thus provide a signal boost compared to using either one alone..”, we investigated this by incubating the tissue sample in a mixture of pGk5b mixed with a farnesylated version followed by anchoring, gelation, digestion, and expansion in **Supp. Fig. 6a-b**. However, upon qualitative examination, we did not observe any signal enhancement. We have added this result to **Supp. Fig. 6c**.

Line 162, and supplementary Fig 8: the mitochondria are stained intensely. Some of these signals could be from endogenously biotinylated proteins that are commonly found on mitochondria (<https://doi.org/10.1385/1-59745-579-2:111>). This can be a good point to mention when the authors replaced biotin with azide (Line 179).

24. Regarding “Line 162, and supplementary Fig 8: the mitochondria are stained intensely. Some of these signals could be from endogenously biotinylated proteins that are commonly found on mitochondria (<https://doi.org/10.1385/1-59745-579-2:111>). This can be a good point to mention when the authors replaced biotin with azide (Line 179).” - we have added a sentence to this end, to the manuscript.

Line 171, and supplementary Fig. 9: perhaps a better mouse line to use is membrane-EGFP mouse line (<https://www.jax.org/strain/007676>). However, this is relatively a minor point.

25. Regarding “Line 171, and supplementary Fig. 9: perhaps a better mouse line to use is membrane-EGFP mouse line (<https://www.jax.org/strain/007676>). However, this is relatively a minor point.”, **Supp. Fig. 9** uses Thy1-YFP mice, which have been used in countless studies to fill cells with YFP, including in ExM contexts. Indeed, in previous collaborative work from our lab, Thy1-YFP was shown to fill cells up to known markers of membranes (e.g., organelle boundaries, synaptic termini, etc.; *Science* 363(6424):eaau8302), so we focused our efforts on these mice.

Line 181, and supplementary Fig 10:

(1) Again, the bars in (c) are not very helpful.

26. Regarding “Again, the bars in (c) are not very helpful.”, we changed **Supp. Fig. 10c** to a box graph as suggested by the reviewer above.

(2) Have the authors explored other handles, such as small peptide tags? This can perhaps be in the discussion section for future developments.

27. Regarding “Have the authors explored other handles, such as small peptide tags? This can perhaps be in the discussion section for future developments.”, we considered biotin and azide because they are very small handles commonly employed for selective and efficient labeling. Indeed, small peptide tags like the HA tag are an option. But, it's uncertain what advantages they would offer, compared to azide and biotin. They would be bigger in size, potentially hampering diffusion, and their interactions with targets might be lower-affinity than the extremely powerful, well-characterized, selective, and efficient click and biotin-streptavidin reactions.

(3) DBCO itself is lipophilic and can contribute to the membrane signal. It would be helpful to conduct a control experiment in which only DBCO-dye is used.

28. Regarding “DBCO itself is lipophilic and can contribute to the membrane signal. It would be helpful to conduct a control experiment in which only DBCO-dye is used,”: we performed a control experiment as the reviewer suggested. In particular, we performed umExM but without the pGk13a staining step. We observed that umExM without the pGk13a staining step provided almost negligible signals, when images were taken under identical optical settings, and processed the same way, as umExM images (**Supp. Fig. 12**).

Line 197 and supplementary Fig 11: the bars should be avoided in these graphs (11c).

29. Regarding “Line 197 and supplementary Fig 11: the bars should be avoided in these graphs (11c).”, we changed **Supp. Fig. 11c** to a box graph as suggested by the reviewer above.

Line 236-239, and Figure 2q-r: these are motile cilia (about a dozen per cell on ependymal cells and airway epithelium, etc) and not primary cilia (one per cell, found on neurons, beta cells in the pancreas, among many others). The text in 2r would need to be changed. In addition, the references are for primary cilia (38-30) and need to be replaced (for an example, <https://doi.org/10.3109/2000-1967-010>).

30. Regarding “Line 236-239, and Figure 2q-r: these are motile cilia (about a dozen per cell on ependymal cells and airway epithelium, etc) and not primary cilia (one per cell, found on neurons, beta cells in the pancreas, among many others). The text in 2r would need to be changed. In addition, the references are for primary cilia (38-30) and need to be replaced (for an example, <https://doi.org/10.3109/2000-1967-010>)”, we thank the reviewer for pointing this out. As the reviewer suggested, text as well as citations were updated accordingly.

n-r and v: the bars are not helpful for visualization.

31. Regarding “n-r and v: the bars are not helpful for visualization.”, we changed **Fig. 2.n-r and v** to a box graph as suggested by the reviewer above.

Line 243, and supplementary Fig 13: choroid plexus cells also have primary cilia (<https://doi.org/10.1016/j.devcel.2023.10.003>). Just curious - did the authors see these in their datasets?

32. Regarding “Line 243, and supplementary Fig 13: choroid plexus cells also have primary cilia (<https://doi.org/10.1016/j.devcel.2023.10.003>). Just curious - did the authors see these in their datasets?”, we only imaged a few regions from each sample, so it’s possible that we only saw the motile cilia; we were not searching for primary cilia and thus may have missed them.

Line 254: throughout the text, the authors used laser exposure time as an indication of the amount of the laser excitation/light dose used. Without knowing the laser excitation power (mW), however, these numbers lack context. Would be helpful to establish a baseline by standard samples, for example, a cell expresses regular cytoplasmic EGFP.

33. Regarding “throughout the text, the authors used laser exposure time as an indication of the amount of the laser excitation/light dose used. Without knowing the laser excitation power (mW), however, these numbers lack context. Would be helpful to establish a baseline by standard samples, for example, a cell expresses regular cytoplasmic EGFP.”, we measured the laser excitation power (mW) to provide additional context to the readers as suggested by the reviewer. To do so, we employed a Nikon W1 spinning disk equipped with a four-line laser system, which we used for imaging throughout the manuscript. Since we utilized the 561nm laser line for pGk13a signals, and reported exposure time for this line, we measured the laser excitation power (mW) of this line. This measurement was performed using a power meter to directly measure the excitation light output from the 4x, 10x, 40x and 60x objective lenses that were used for imaging throughout the manuscript. We have added this in supplementary table 4:

Lens	Laser power	
	50%	100%
4x	1.10 mW	2.33mW
10x	1.11mW	2.35mW
40x	0.97mW	2.02mW
60x	0.94mW	2.00mW

Line 257 and the remainder of the paragraph: related to the statement above - was the same laser power used?

34. Regarding “Line 257 and the remainder of the paragraph: related to the statement above - was the same laser power used?”, yes, the same laser power was used for imaging.

Line 275, and Fig. 2t-v: it would be helpful to point out that these are motile cilia (see comment #13 above).

35. Regarding “Line 275, and Fig. 2t-v: it would be helpful to point out that these are motile cilia (see comment #13 above).”, we have updated text and citations as the reviewer suggested.

Line 282, and Fig 2v: can the authors comment a bit more in regards to the context of a gap of 60 nm?

36. Regarding “Line 282, and Fig 2v: can the authors comment a bit more in regards to the context of a gap of 60 nm?”, we used a gap of 60nm since the effective resolution of umExM is ~60nm (**Fig. 2g**). This was explained in the figure legend of **Fig. 2v** in the initial manuscript.

Line 350: Fig 4e should be mentioned here in the text. The bars in the graph are not that helpful (see also comments above).

37. Regarding “Fig 4e should be mentioned here in the text. The bars in the graph are not that helpful (see also comments above).”, we added text to mention **Fig. 4e**, and we changed **Fig. 4e** to a box graph as suggested by the reviewer above.

Line 395: should this modified protocol be used in other areas of the brain as well? A brief mention of this would suffice.

38. Regarding “should this modified protocol be used in other areas of the brain as well? A brief mention of this would suffice.”: the modified protocol works for cortex, hippocampus, 3rd ventricle, choroid plexus as well as the corpus callosum. However, the modified protocol requires additional time and resources (i.e., necessitating two gels) compared to the unmodified protocol, which is suitable for all these regions except the corpus callosum. Hence, we suggest employing the modified protocol specifically for the corpus callosum region. We added a sentence regarding this in the legend of Supp. Fig. 23.

Reviewer #3 (Remarks to the Author):

The manuscript introduces ultrastructural membrane expansion microscopy (umExM), a novel technique for high-resolution imaging of cellular membranes. The authors achieve high label density through a multi-faceted approach centered on the development and application of novel synthetic amphiphilic membrane labeling probes. These probes are designed to densely and uniformly label membrane structures, including plasma membranes and various organelle membranes, within fixed tissues. The core ideas behind achieving such high label density include:

- The membrane labeling probes exhibit lipophilicity, akin to traditional fluorescent lipophilic dyes. This characteristic enables the probes to preferentially localize and diffuse within membrane structures, ensuring comprehensive membrane coverage.
- The probes are designed with a chemical handle that allows for the selective attachment of fluorophores after the formation of the ExM polymer network. This feature ensures that the

labeling probes remain small, facilitating their diffusion within the tissue and preserving the fluorophore's integrity during the polymerization process.

- The probes possess a polymer-anchorable handle, enabling their incorporation into the swellable polymer network created during the ExM process. This integration is crucial for the physical expansion of the labeled membranes alongside the hydrogel, maintaining the high density of labeling through the expansion process.
- Through systematic design and optimization, including the selection of the hydrocarbon side chains and the number of lysines in the probe backbone, the authors tailored the probes for optimal performance. This optimization process involved exploring the chemical space to identify the probe configurations that best support dense membrane staining with minimal distortion and high fidelity.
- The ExM protocol itself is optimized to accommodate these novel probes, including adjustments to the chemical softening of the sample, the temperature control during processing, and the specific conditions for hydrogel formation and expansion. These adjustments ensure the preservation of membrane integrity and the effectiveness of labeling throughout the ExM process.

The authors apply their novel methodology to fixed mouse brain tissue. They specifically focus on labeling and visualizing various membranous structures within this tissue, including:

- Plasma Membranes: The outer membranes of cells that act as the boundary between the cell interior and the external environment.
- Mitochondrial Membranes: The membranes enclosing mitochondria, the cell's powerhouse, involved in energy production.
- Nuclear Membranes: The double membrane structure surrounding the nucleus, where genetic material is stored.
- Ciliary Membranes: The membrane of cilia, which are small hair-like structures that protrude from the cell body and are involved in movement and sensing the external environment.
- Myelin Sheaths: The fatty membrane that ensheathes and insulates axons, enhancing the speed of electrical signal transmission in the nervous system.
- Extracellular Vesicle Membranes: The membranes surrounding extracellular vesicles, which are small particles released from cells that can carry signaling molecules to other cells.

By applying umExM to these diverse membrane systems, the authors are able to achieve dense and continuous labeling with high signal-to-background ratios, enabling the visualization of membranous structures at a resolution of approximately 60 nm. This methodology supports co-visualization of membranous structures alongside proteins and RNAs, allowing for the segmentation of neuronal compartments such as cell bodies, dendrites, and axons, and tracing of neuronal processes.

The manuscript is well-structured and logically coherent, effectively conveying the significance and innovation of umExM. I could not find any linguistic or logical errors. Therefore, I recommend publication of the manuscript as is.

39. We appreciate the reviewer's thoughtful comments and positive feedback.

References

1. Wang, U.-T. T. *et al.* Protein and lipid expansion microscopy with trypsin and tyramide signal amplification for 3D imaging. *Sci. Rep.* **13**, 21922 (2023).
2. Damstra, H. G. J. *et al.* Visualizing cellular and tissue ultrastructure using Ten-fold Robust Expansion Microscopy (TREx). *Elife* **11**, (2022).
3. Nijenhuis, W. *et al.* Optical nanoscopy reveals SARS-CoV-2-induced remodeling of human airway cells. *bioRxiv* 2021.08.05.455126 (2021).
4. Götz, R. *et al.* Nanoscale imaging of bacterial infections by sphingolipid expansion microscopy. *Nat. Commun.* **11**, 6173 (2020).
5. M'Saad, O. *et al.* All-optical visualization of specific molecules in the ultrastructural context of brain tissue. *bioRxiv* 2022.04.04.486901 (2022).
6. Tillberg, P. W. *et al.* Protein-retention expansion microscopy of cells and tissues labeled using standard fluorescent proteins and antibodies. *Nature Biotechnology* **2016** 34:9 **34**, 987–992 (2016).
7. Zhao, W. *et al.* Enhanced detection of fluorescence fluctuations for high-throughput super-resolution imaging. *Nat. Photonics* 1–8 (2023).
8. Culley, S. *et al.* Quantitative mapping and minimization of super-resolution optical imaging artifacts. *Nat. Methods* **15**, 263–266 (2018).
9. Chang, J. B. *et al.* Iterative expansion microscopy. *Nature Methods* **2017** 14:6 **14**, 593–599 (2017).
10. Schmidt, R. *et al.* Mitochondrial cristae revealed with focused light. *Nano Lett.* **9**, 2508–2510 (2009).
11. Kuan, A. T. *et al.* Dense neuronal reconstruction through X-ray holographic nanotomography. *Nat. Neurosci.* **23**, 1637 (2020).

12. Skepper, J. N. Immunocytochemical strategies for electron microscopy: choice or compromise. *J. Microsc.* **199**, 1–36 (2000).
13. Eltoun, I., Fredenburgh, J., Myers, R. B. & Grizzle, W. E. Introduction to the Theory and Practice of Fixation of Tissues. *J. Histotechnol.* **24**, 173–190 (2001).
14. Hobot, J. A. & Newman, G. R. Strategies for improving the cytochemical and immunocytochemical sensitivity of ultrastructurally well-preserved, resin embedded biological tissue for light and electron microscopy. *Scanning Microsc. Suppl.* **5**, S27-40; discussion S40-1 (1991).
15. Palay, S. L., McGEE-RUSSELL, S. M., Gordon, S. & Grillo, M. A. FIXATION OF NEURAL TISSUES FOR ELECTRON MICROSCOPY BY PERFUSION WITH SOLUTIONS OF OSMIUM TETROXIDE. *J. Cell Biol.* **12**, 385–410 (1962).
16. Gambarotto, D. *et al.* Imaging cellular ultrastructures using expansion microscopy (U-ExM). *Nat. Methods* **16**, 71–74 (2019).
17. Nusser, Z. *et al.* Cell type and pathway dependence of synaptic AMPA receptor number and variability in the hippocampus. *Neuron* **21**, 545–559 (1998).
18. Eyre, M. D. & Nusser, Z. Only a Minority of the Inhibitory Inputs to Cerebellar Golgi Cells Originates from Local GABAergic Cells. *eNeuro* **3**, 317–334 (2016).
19. Glaser, A. *et al.* Expansion-assisted selective plane illumination microscopy for nanoscale imaging of centimeter-scale tissues. *bioRxiv* (2023) doi:10.1101/2023.06.08.544277.
20. Dunn, A. K., Smithpeter, C., Welch, A. J. & Richards-Kortum, R. Sources of contrast in confocal reflectance imaging. *Appl. Opt.* **35**, 3441–3446 (1996).
21. Chen, F., Tillberg, P. W. & Boyden, E. S. Expansion microscopy. *Science* **347**, 543–548 (2015).

We have now responded to all the reviewers' comments.

REVIEWERS' COMMENTS

Reviewer #1 (Remarks to the Author):

I would like to thank the authors for their work and for answering my questions so thoroughly. Overall this manuscript represents a high-quality contribution to the field, with an impressive range of controls and analyses that strengthen the validity of the results. I have no further comments to make and fully support its publication.

1. We thank the reviewer for their appreciative comments.

Reviewer #2 (Remarks to the Author):

The authors had done a great job responding to my previous comments in the revised manuscript. I only have one very minor comment:

In Supplementary Figure 12 legend (Line 433-435):

a) Representative (n=2 fixed brain slices from two mice) single z-plane confocal image of expanded mouse brain tissue (CA1), that underwent the umExM protocol, but without pGk13a staining of the membrane, showing DBCO-Cy3 staining of the membrane (inverted gray).

Since in non-pGk12b conditions, barely any signal was present. Therefore, I believe a better description would be:

a) Representative (n=2 fixed brain slices from two mice) single z-plane confocal image of expanded mouse brain tissue (CA1), that underwent the umExM protocol, but without pGk13a staining of the membrane, showed negligible signals (inverted gray).

The manuscript is otherwise beautifully written with wonderful results, and will be very helpful to advance of the field in ExM and thus potentially uncover new biology.

2. We thank the reviewer for their appreciative comments. As suggested, we have updated the legend for Supplementary Figure 12, which is now Supplementary Figure 14.